# Domain-Inspired Sharpness-Aware Minimization Under Domain Shifts

**Ruipeng Zhang[†,‡], Ziqing Fan[†,‡], Jiangchao Yao[†,‡,✉], Ya Zhang[†,‡], Yanfeng Wang[†,‡,✉]**
[†] Cooperative Medianet Innovation Center, Shanghai Jiao Tong University
[‡] Shanghai Artificial Intelligence Laboratory
{zhangrp, zqfan_knight, Sunarker, ya_zhang, wangyanfeng}@sjtu.edu.cn

## Abstract

This paper presents a Domain-Inspired Sharpness-Aware Minimization (DISAM) algorithm for optimization under domain shifts. It is motivated by the inconsistent convergence degree of SAM across different domains, which induces optimization bias towards certain domains and thus impairs the overall convergence. To address this issue, we consider the domain-level convergence consistency in the sharpness estimation to prevent the overwhelming (deficient) perturbations for less (well) optimized domains. Specifically, DISAM introduces the constraint of minimizing variance in the domain loss, which allows the elastic gradient calibration in perturbation generation: when one domain is optimized above the averaging level *w.r.t.* loss, the gradient perturbation towards that domain will be weakened automatically, and vice versa. Under this mechanism, we theoretically show that DISAM can achieve faster overall convergence and improved generalization in principle when inconsistent convergence emerges. Extensive experiments on various domain generalization benchmarks show the superiority of DISAM over a range of state-of-the-art methods. Furthermore, we show the superior efficiency of DISAM in parameter-efficient fine-tuning combined with the pretraining models. The source code is released at https://github.com/MediaBrain-SJTU/DISAM.

## 1 Introduction

Although deep learning has achieved remarkable advances in various areas (He et al., 2016; Dosovitskiy et al., 2020), it remains a challenge for optimization in pursuit of strong generalization. Especially, a lower training loss does not necessarily guarantee a better generalization, as there exist numerous local minima in the complex and non-convex hypothesis space. Recent empirical and theoretical investigations (Dziugaite & Roy, 2017; Chaudhari et al., 2019; Jiang et al., 2020; 2023; Dinh et al., 2017b; Keskar et al., 2017b) have identified a significant correlation between generalization and the sharpness of the loss landscape. This correlation suggests that generalizability can be interpreted as flatness in the loss surface, leading to a wide range of explorations that have contributed to the rapid development of Sharpness-Aware Minimization (SAM) (Foret et al., 2021).

Existing SAM-based methods predominantly focus on the narrowly defined generalizability between training and test data under the Independent and Identically Distributed (i.i.d) assumption, which can be summarized as two categories. The first strives to improve the performance by creating a more effective estimation of sharpness like the enhanced minimization in GSAM (Zhuang et al., 2022), PGN (Zhao et al., 2022), SAGM (Wang et al., 2023b) and VaSSO (Li & Giannakis, 2023), as vanilla perturbation in SAM fails to accurately capture the geometric flatness of the loss landscape. The other category targets to improve computational efficiency by reducing perturbation directions (Liu et al., 2022) or using a more efficient perturbation surrogate (Du et al., 2022a;b), as the original SAM incurs double the computational overhead compared to Empirical Risk Minimization (ERM). Nonetheless, these methods cannot solve generalizability scenarios that involve training data of multiple domains with domain shifts like *Domain Generalization (DG)* (Ben-David et al., 2010; Li et al., 2017).

In this study, we observed that sometimes SAM even has a detrimental impact in situations where there exist domain shifts across multiple domains as shown in Figure 1(a). While a few studies have incorporated SAM-based methods in domain generalization tasks (Wang et al., 2023b; Foret et al.,

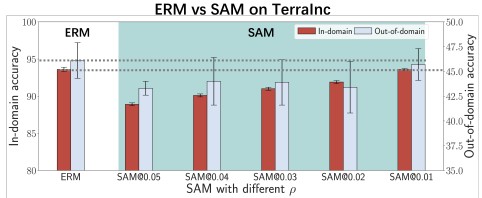

(a) Performance under domain shifts.

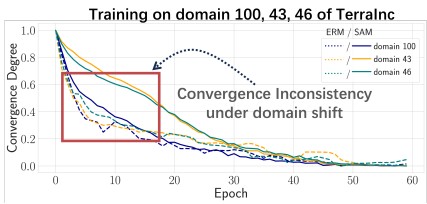

(b) Convergence curves under domain shifts.

Figure 1: Illustration of SAM's degradation of the training process under domain shifts. (a) Performance comparison between ERM and SAM, where SAM consistently performs worse than ERM across all hyperparameters $\rho$. (b) Convergence curves of SAM and ERM for each domain during training, with the convergence degree normalized to [0,1]. SAM exacerbates the disparity in convergence degree among different domains in domain shift scenarios, resulting in inferior generalization performance. The dataset used here is `TerraInc` from the DomainBed benchmark, and the backbone is ResNet50. Further experimental details are provided in Section 4.1 and Appendix C.5.

2021), they cannot ensure consistent improvements in generalizability during domain shifts due to their reliance on the i.i.d assumption. Upon a thorough analysis of the behavior of SAM under domain shifts, we discovered that the degradation of the training process caused by SAM from the disparity in convergence degree among different domains as shown in Figure 1(a). Given the inconsistency in the degree and direction of convergence among different domains during training (Arjovsky et al., 2019; Krueger et al., 2021), the straightforward application of SAM for perturbations may not only disrupt convergence but also generate perturbation directions that are not adequately coherent to the geometric characteristics of the entire loss landscape.

To solve the aforementioned problem, we propose a Domain-Inspired Sharpness-Aware Minimization (DISAM) algorithm. As the degradation origins from the inconsistency in convergences of these domains, DISAM incorporates domain-level convergence information to intervene in the perturbation of the vanilla SAM: *the perturbation direction should focus more on domains with higher convergence degree while being mild to domains with lower convergence degree.* Under such balancing in perturbation, the gradient update actually implements the domain-level personalization, thus mitigates the impact of domain shifts and enhances the generalization performance. Technically, we ingeniously accomplish the adaptive adjustment of the perturbation direction in accordance with the degree of convergence through the domain loss variance minimization constraint. The perturbation of DISAM directs towards a location with a more consistent convergence degree, enabling a better global view of the loss landscape for gradient update. We summarize our contributions as follows:

- We identify that the use of SAM has a detrimental impact on training under domain shifts, thereby compromising generalizability, and further analyze that the reason is the inconsistent convergence of training domains that deviates from the underlying i.i.d assumption of SAM.

- We introduce a novel approach called Domain-Inspired Sharpness-Aware Minimization to mitigate the problem above. DISAM incorporates domain-level convergence consistency by imposing a variance minimization constraint on domain loss during the sharpness estimation process, thereby enabling a more representative perturbation location and enhancing generalization.

- Extensive experiments show the superiority of DISAM in improving the current state-of-the-art methods on several benchmarks. We also provide a comprehensive analysis of its merit of faster convergence compared to SAM, and show its persistent generalization capabilities under parameter-efficient fine-tuning with large models like CLIP.

## 2 PRELIMINARIES

### 2.1 BASIC NOTATIONS

- $\mathcal{S} = \{D_1, D_2, \cdots, D_M\}$: Overall training set of a $M$-source domain generalization task. We denote each domain by $D_i$ and the number of samples in $D_i$ by $n_i = |D_i|$.
- $\xi, \xi_j^i$: A specific sample and the j-th sample in i-th domain $D_i$, respectively.

- $\mathcal{L}$, $\mathcal{L}(w)$, $\mathcal{L}(w; \xi)$: A loss function, expected loss under $w$ and the specific loss of $\xi$, respectively.
- $\mathcal{L}_i(w) = \mathbb{E}_{\xi \in D_i} \mathcal{L}(w; \xi)$: Expected loss under $w$ for each domain $D_i$.
- $\text{Var}\{\cdot\}_{i=1}^M$: The variance among $M$ training domains, which holds: $\text{Var}\{\mathcal{L}_i(w)\}_{i=1}^M = \frac{1}{2M^2} \sum_{i=1}^M \sum_{j=1}^M (\mathcal{L}_i(w) - \mathcal{L}_j(w))^2$.
- $\mathcal{L}_{DI}(w) = \mathcal{L}(w) - \lambda \text{Var}\{\mathcal{L}_i(w)\}_{i=1}^M$: A loss function with domain-inspired regularizer $\text{Var}\{\mathcal{L}_i(w)\}_{i=1}^M$ on $\mathcal{L}$. $\lambda$ is a constant value that controls the strength of the constraint.
- $\mathcal{L}_p(w) = \max_{\|\epsilon\|_2 \le \rho} \mathcal{L}(w + \epsilon)$: The perturbed loss and the objective of SAM.
- $w_t^{asc} = w_t + \rho \frac{\nabla \mathcal{L}_{DI}(w_t)}{\|\nabla \mathcal{L}_{DI}(w_t)\|}$: The sharpness estimation of DISAM with gradient ascend at step $t$.
- $\eta_t$: Learning rate at step $t$.
- $w$: Parameters of a neural network $\in \mathbb{R}^k$, where $k$ is the dimension.
- $\epsilon \in \mathbb{R}^k$: A perturbation on the parameters $w$ with scale $\rho \in \mathbb{R}$.

## 2.2 SHARPNESS-AWARE MINIMIZATION

In general, simply minimizing ERM tends to overfit training data and extensive studies show the correlation between generalizability and the sharpness of minima (Dinh et al., 2017b; Hochreiter & Schmidhuber, 1994b; McAllester, 1999; Chaudhari et al., 2019). We clarify the concepts as below.

**Sharpness.** The *sharpness* on parameter $w$ with a dataset $D$ and loss function $\mathcal{L}$ is:

$$s(w, D) \triangleq \max_{\|\epsilon\|_2 \le \rho} \mathbb{E}_{\xi \in D}[\mathcal{L}(w + \epsilon; \xi) - \mathcal{L}(w; \xi)]. \tag{1}$$

**Sharpness-Aware Minimization (SAM).** Foret et al. (2021) proposed SAM to improve the generalization by simultaneously minimizing the loss and the sharpness of the overall loss surface. The objective is defined as:

$$\min_w \max_{\|\epsilon\|_2 \le \rho} \mathbb{E}_{\xi \in D}[\mathcal{L}(w + \epsilon; \xi)]. \tag{2}$$

From the above equation, we can see SAM minimizes a perturbed loss "$\max_{\|\epsilon\|_2 \le \rho} \mathbb{E}_{\xi \in D}[\mathcal{L}(w + \epsilon; \xi)]$", which aims to maximize the loss $\mathcal{L}$ within radius $\rho$ centered at the parameter $w$.

## 3 METHOD

### 3.1 MOTIVATION

Although existing SAM-based methods that minimize the sharpness have achieved good generalization, in the case of multiple domains with shifts, the inherent heterogeneity in quantity and task difficulty among domains can considerably distort their sharpness estimation, yielding a degradation in the performance. Concretely, with a collection $\mathcal{S}$ of $M$ domains, each of which contains a set of $n_i$ samples, *i.e.,* $\{\xi_j^i = (x_j^i, y_j^i)\}_{j=1}^{n_i}$, the training objective can be then formulated as follows:

$$\min_w \mathbb{E}_{\xi \in \mathcal{S}}[\mathcal{L}(w; \xi)] = \frac{1}{N} \sum_{i=1}^M \sum_{j=1}^{n_i} \mathcal{L}(w; \xi_j^i) = \sum_{i=1}^M \alpha_i \mathcal{L}_i(w), \tag{3}$$

where $N = \sum_{i=1}^M n_i$, $\alpha_i = \frac{n_i}{N}$ and $\mathcal{L}_i(w) = \frac{1}{n_i} \sum_{j=1}^{n_i} \mathcal{L}(w; \xi_j^i)$. Note that, we clarify here that we will ignore the notations of data properly in some subsequent equations to avoid clutter. Then, on the basis of Eq. (3), the corresponding objective of SAM under domain shifts is defined as:

$$\min_w \mathbb{E}_{\xi \in \mathcal{S}}[\mathcal{L}_{SAM}(w; \xi)] = \min_w \max_{\|\epsilon\|_2 \le \rho} \mathbb{E}_{\xi \in \mathcal{S}}[\mathcal{L}(w + \epsilon; \xi)] \stackrel{?}{\approx} \min_w \max_{\|\epsilon\|_2 \le \rho} \sum_{i=1}^M \alpha_i \mathcal{L}_i(w + \epsilon). \tag{4}$$

The core that we should point out is whether the approximation from $\max_{\|\epsilon\|_2 \le \rho} \mathbb{E}_{\xi \in \mathcal{S}}[\mathcal{L}(w + \epsilon; \xi)]$ to $\max_{\|\epsilon\|_2 \le \rho} \sum_{i=1}^M \alpha_i \mathcal{L}_i(w + \epsilon)$ in Eq. (4) is reasonable. There is no harm when samples in $\mathcal{S}$

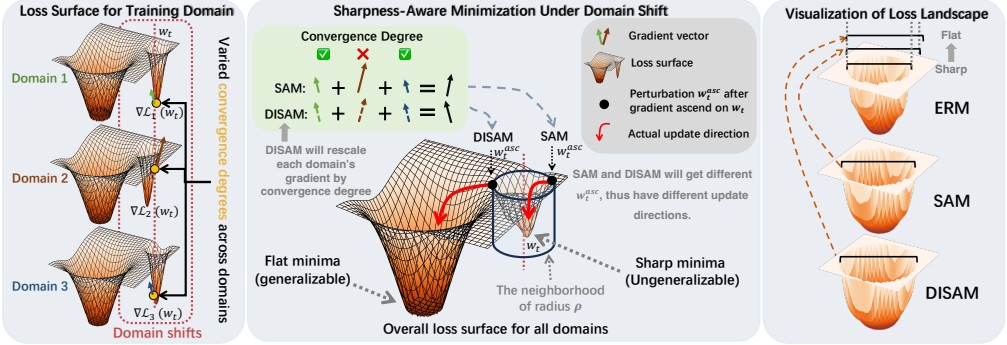

Figure 2: Toy example illustrating the problem of **SAM under domain shifts**. *Left:* Domain shifts on the loss surface of training domains, which causes the inconsistency of convergence degree. *Middle:* Differences between SAM and DISAM in the perturbation generation and convergence. Specifically, SAM is affected by the inconsistent degree of convergence. *Right:* Visualization of loss landscape for ERM, SAM, and DISAM on unseen test domain. DISAM is flatter than SAM and ERM.

are intrinsically independently and identically distributed. However, this is actually ill-posed under domain shifts. Differences in the amount of data or inconsistency in task difficulty can result in biased sharpness estimation towards specific domains, hindering the overall convergence. As shown in Figure 2, neglecting the domain shifts and the consequent convergence inconsistency issue, using SAM directly at this point, significantly misdirects the perturbation direction towards the domain with the largest gradient vectors (implying a lower degree of convergence). Consequently, it does not help find a better convergence path and conversely leads to a suboptimal sharp minima.

## 3.2 Domain-Inspired SAM

To address the problem described in Eq. (4) and Figure 2, we need to design an adjustment mechanism that takes into account the convergence degree of each domain during the perturbation generation. Specifically, we should make the perturbation direction $\nabla \mathcal{L}_p(w)$ to efficiently pull domains that are close to convergence out of sharp minima while minimizing the negative impact on domains that have not yet converged. Here, we define the convergence degree of domain $i$ with model parameter $w$ as $C_i(w) = \mathcal{L}_i^* - \mathcal{L}_i(w)$, where $\mathcal{L}_i^*$ represents the optimal minimum of domain $i$ ($\mathcal{L}_i^* \geq 0$). The design principle is to prioritize the contribution of domains with larger $C_i(w)$ to the overall perturbation direction. To achieve this, a simple approach involves directly adding $C_i(w)$ to the weight $\alpha_i$ of SAM with a controlling coefficient term $\beta$.

$$\sum_{i=1}^{M} \alpha_i \nabla \mathcal{L}_i(w) \rightarrow \sum_{i=1}^{M} \frac{\alpha_i + \beta C_i(w)}{\sum_{j=1}^{M}(\alpha_j + \beta C_j(w))} \nabla \mathcal{L}_i(w) = \sum_{i=1}^{M} \frac{\beta(C_i(w) - \alpha_i \sum_{j=1}^{M} C_j(w))}{1 + \beta \sum_{j=1}^{M} C_j(w)} \nabla \mathcal{L}_i(w) \quad (5)$$

However, we observe that the weight adjustment in Eq. (5) that is affected by the convergence degree, is constrained by the magnitude of $\alpha_i$. That is, domains with higher $\alpha_i$ values can tolerate lower convergence degrees, which may not accurately satisfy our goals. To refine this, we propose to use an adaptive way to ensure fairness by calculating the average convergence at the domain level, namely, $C_i(w) - \alpha_i \sum_{i=1}^{M} C_i(w) \rightarrow \mathcal{L}_i(w) - \frac{1}{M} \sum_{i=1}^{M} \mathcal{L}_i(w)$. With this intuition, we introduce a method called *Domain-Inspired Sharpness-Aware Minimization (DISAM)* that incorporates a variance constraint between domain losses to estimate sharpness. It enables the adaptive adjustment of the perturbation direction similar to our spirit, which we will provide a detailed analysis in the following Eq. (8). First of all, we give the definition of the variance between different domain losses as:

$$\text{Var}\{\mathcal{L}_i(w + \epsilon)\}_{i=1}^{M} = \frac{1}{2M^2} \sum_{i=1}^{M} \sum_{j=1}^{M} (\mathcal{L}_i(w + \epsilon) - \mathcal{L}_j(w + \epsilon))^2. \quad (6)$$

Then, putting the above variance term into the loss, the new training objective can be defined as:

$$\min_{w} \mathbb{E}_{\xi \in \mathcal{S}}[\mathcal{L}_{DISAM}(w; \xi)] \triangleq \min_{w} \max_{\|\epsilon\|_2 \leq \rho} \left[ \sum_{i=1}^{M} \alpha_i \mathcal{L}_i(w + \epsilon) - \lambda \text{Var}\{\mathcal{L}_i(\hat{w} + \epsilon)\}_{i=1}^{M} \right] \quad (7)$$

Here $\hat{w}$ is $w$ without derivative taken during backpropagation, and it only makes effect in the $\max_{\|\epsilon\|_2 \leq \rho}$ loop without affecting the optimization of the first term *w.r.t.* $w$. Following the computing way of perturbation $\epsilon$ in SAM, namely, using first-order Taylor expansion (Foret et al., 2021), we will have $\epsilon \approx \rho \frac{\nabla \mathcal{L}_{DISAM}}{\|\nabla \mathcal{L}_{DISAM}\|}$, where $\nabla \mathcal{L}_{DISAM}$ *w.r.t.* $w$ has the form:

$$
\begin{aligned}
\nabla \mathcal{L}_{DISAM} &= \sum_{i=1}^{M} \left( \alpha_i - \frac{2\lambda}{M} \left( \mathcal{L}_i(w) - \frac{1}{M} \sum_{j=1}^{M} \mathcal{L}_j(w) \right) \right) \nabla \mathcal{L}_i(w) \\
&= \nabla \mathcal{L}_{SAM} - \underbrace{\sum_{i=1}^{M} \frac{2\lambda}{M} \left( \mathcal{L}_i(w) - \frac{1}{M} \sum_{j=1}^{M} \mathcal{L}_j(w) \right) \nabla \mathcal{L}_i(w)}
\end{aligned}
\tag{8}
$$

**Adaptive adjustment:** *increase* weights for smaller losses, *reduce* for larger ones.

The first term in the RHS of Eq. (8) recovers the gradient term for perturbation generation in SAM, and the second term characterizes the working mechanism for the adaptive adjustment. As can be seen, when the loss $\mathcal{L}_i(w)$ of one certain domain is above the averaging level, the second term will generate a residual gradient for this domain to cancel out the gradient contribution in $\nabla \mathcal{L}_{SAM}$, and vice versa. It means to have a mild perturbation for the domain that is not well optimized, and have an aggressive perturbation for the domain that is well optimized. In total, the variance constraint ensures that the perturbation location is at a more consistent point, enabling a better global view of the loss landscape for gradient update. The complete algorithm is described in Appendix B. Regarding $\lambda$, a default value of $0.1$ is relatively stable, and we provide more discussion about $\lambda$ in Appendix B.3.

**Difference and Compatibility.** Similarly, the current SAM variants will meet the same challenge, if they are directly applied to this scenario. Different from existing state-of-the-art methods like GSAM (Zhuang et al., 2022) and SAGM (Wang et al., 2023b) that modify the optimization objective based on the second derivative of SAM, DISAM rectifies the domain shift issue by the domain-level adjustment in the perturbation generation, which actually alleviates the negative impacts on the training objective. In Appendix A.2.5, we present a table to comprehensively characterize the difference between DISAM and other domain-invariant robust optimization methods. Besides, DISAM can be easily extended into other SAM-based methods to improve the generalization performance. We have provided a comparison of the similarities and differences between DISAM and general convergence consistency methods (such as V-REx(Krueger et al., 2021) and Fishr(Rame et al., 2022)) in Appendix B.1.

### 3.3 UNDERSTANDING DOMAIN-INSPIRED SAM

**Complexity.** Compared to SAM-based methods, our algorithm only additionally computes the loss variance between different domains as Eq. (6) and requires no extra storing space. Therefore it has the same space complexity and the time complexity can be represented as $O_{\text{DISAM}} = O_{\text{SAM}} + O_{\text{Var}}$. Since it only needs to additionally count the domain loss and the corresponding variance according to the domain label when calculating the empirical loss, the overall cost on $O_{\text{Var}}$ is negligible.

**Convergence.** In the following, we provide the convergence analysis of SAM and DISAM. Similar to (Zhuang et al., 2022; Jiang et al., 2023), our theorem is established on assumptions that a non-convex function $\mathcal{L}(w)$ is $L$ Lipschitz-smooth, the lower bound of the empirical loss is bounded by $\mathcal{L}_{min}$, and the norm of noisy stochastic gradients is bounded ($\|\nabla \mathcal{L}_p(w_t)\|_2 \leq G$) at the t-step.

**Theorem 1.** *Consider a non-convex function $\mathcal{L}(w)$ with Lipschitz-smooth constant $L$ and lower bound $\mathcal{L}_{min}$. With the bounded norm assumption of noisy stochastic gradients ($\|\nabla \mathcal{L}_p(w)\|_2 \leq G$) at the t-step, the learning rate $\eta_t = \eta_0 / \sqrt{t}$ and a fixed perturbation amplitude $\rho$, we have:*

$$
\frac{1}{T} \sum_{t=1}^{T} \mathbb{E} \|\nabla \mathcal{L}_p(w_t)\|_2^2 \leq \frac{\mathcal{L}_p(w_0) - \mathcal{L}_{min}}{\eta_0} \frac{1}{\sqrt{T}} + \frac{(LG^2 + \rho^2 L \Gamma^2) \eta_0 \log(T)}{\sqrt{T}},
$$

*where in SAM, $\Gamma = L$ and in our DISAM, $\Gamma \leq L$.*

The complete proof is presented in Appendix B. As can be seen in Theorem 1, the critical convergence difference between SAM and DISAM is on $\Gamma$, and especially the $\Gamma$ in our DISAM is smaller than that in SAM due to the canonically correlated perturbations during training (see the proof for the details),

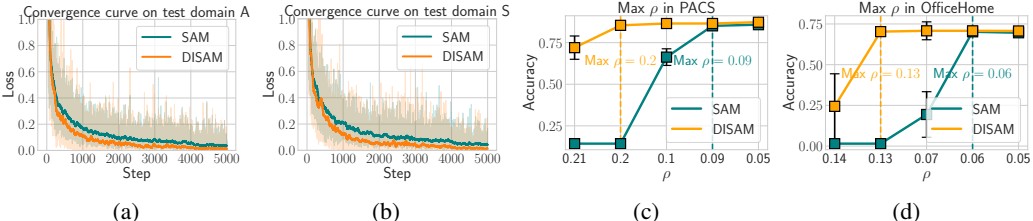

Figure 3: **Convergence curves** and **Max $\rho$ search** for SAM and DISAM. (a) & (b) show the trend of $\mathcal{L}(w)$ during the training process on PACS dataset, while (c) & (d) search for the maximum perturbation amplitude $\rho$ of SAM and DISAM on PACS and OfficeHome datasets.

which thus leads to a faster convergence rate. Note that, the overall $\rho^2 L \Gamma^2$ in Theorem 1 indicates that a larger perturbation amplitude $\rho$ will result in larger upper bound of convergence. However, as analyzed in SAM (Foret et al., 2021), a larger perturbation amplitude $\rho$ has the merit of reaching a smaller upper bound on generalization error. This means that $\rho$ actually has a trade-off between accelerating the convergence and improving the generalization. Fortunately, when in the same overall value of $\rho^2 L \Gamma^2$, as DISAM enjoys a smaller $\Gamma$ than SAM, DISAM can permit potential larger $\rho$ than that in SAM, thus yielding a better generalization (Please refer to Appendix B.4 for more discussion).

To verify the theoretical analysis, we present our empirical results of DISAM and SAM in Figure 3. As shown in Figures 3(a) and 3(b), the training curves on the PACS dataset show that DISAM achieves faster and steeper convergence than SAM. In addition, as DISAM has a smaller $\Gamma$, it is able to utilize a larger perturbation amplitude $\rho$. In Figures 3(c) and 3(d), we show the experimental support that DISAM allows larger $\rho$ values (0.2 and 0.13) than SAM (0.09 and 0.06) on PACS and OfficeHome datasets, while achieving a better performance. In total, these experiments confirm the advantage of DISAM that allows larger $\rho$ for better generalization.

## 4 EXPERIMENTS

### 4.1 EXPERIMENT SETUPS

**Datasets.** We evaluate DISAM on five datasets PACS (Li et al., 2017), VLCS (Fang et al., 2013) OfficeHome (Venkateswara et al., 2017), TerraIncognita (Beery et al., 2018) (abbreviated as TerraInc), and DomainNet (Peng et al., 2019), following the DomainBed benchmark (Gulrajani & Lopez-Paz, 2021). For fair comparison, we adhere to the training and evaluation protocol outlined in DomainBed.

**Evaluation.** The standard leave-one-domain-out strategy is used in evaluation. Specially, the unseen domain is used to evaluate the out-of-domain generalization, and the validation sets of source domains are used to measure the in-domain generalization, while the others are used for training. Final accuracy is averaged across all settings, and the performance is the averaging over three trials with distinct random seeds. Detailed statistics for each case of all datasets are provided in Appendix C.

**Implementation details.** Our backbones are ResNet50 pretrained on ImageNet (He et al., 2016) and a pretrained CLIP (Radford et al., 2021) with ViT-B/16 structure (Dosovitskiy et al., 2020). For model hyperparameters, we adopt settings in (Wang et al., 2023b) for experiments using ResNet50 and in (Shu et al., 2023) for experiments using CLIP. As the default, we set the perturbation hyperparameter $\rho$ to 0.05 (Wang et al., 2023b) (Fixed value during training), and the weight of the variance constraint $\lambda$ to 0.1. For a detailed description of the hyparameter settings, please see Appendix C.

### 4.2 PERFORMANCE UNDER RESNET50 BACKBONE

We propose incorporating our *domain-inspired* adaptive adjustment into three SAM-based methods: SAM (Foret et al., 2021), GSAM (Zhuang et al., 2022), and SAGM (Wang et al., 2023b) on five datasets of DomainBed with ResNet50 backbone. Table 1 shows that our **Domain-Inspired** SAM can mitigate issues arising from SAM's training under domain shifts, by comparing averaged in-domain and out-of-domain performance of leading SAM methods, with and without DISAM. *In-domain results* show domain-inspired perturbations enhance convergence, especially on the TerraInc dataset with substantial domain gaps. In *Out-of-domain results*, DISAM consistently improves generalization,

Table 1: **Comparison with state-of-the-art domain generalization methods based on ResNet50.** In-domain and Out-of-domain accuracies on five datasets from DomainBed.

| Algorithm | PACS | VLCS | OfficeHome | TerraInc | DomainNet | Avg. |
|---|---|---|---|---|---|---|
| *In-domain results* | | | | | | |
| ERM | $96.6 \pm 0.2$ | $84.6 \pm 0.4$ | $84.2 \pm 0.3$ | $93.6 \pm 0.3$ | $67.1 \pm 1.6$ | 85.2 |
| SAM | $97.3 \pm 0.1$ | $84.8 \pm 0.3$ | $85.8 \pm 0.2$ | $88.9 \pm 0.2$ | $68.5 \pm 0.1$ | 85.1 |
| **Domain-Inspired** | $97.8 \pm 0.1$ | $84.4 \pm 0.3$ | $86.3 \pm 0.2$ | $94.8 \pm 0.2$ | $70.2 \pm 0.1$ | 86.7 |
| GSAM | $97.8 \pm 0.2$ | $83.9 \pm 0.2$ | $85.9 \pm 0.2$ | $92.1 \pm 0.2$ | $69.1 \pm 0.1$ | 85.8 |
| **Domain-Inspired** | $97.9 \pm 0.1$ | $\mathbf{85.1} \pm 0.4$ | $86.2 \pm 0.2$ | $94.8 \pm 0.3$ | $70.0 \pm 0.1$ | 86.8 |
| SAGM | $97.6 \pm 0.1$ | $84.6 \pm 0.3$ | $86.1 \pm 0.2$ | $92.0 \pm 0.2$ | $69.2 \pm 0.1$ | 85.9 |
| **Domain-Inspired** | $\mathbf{97.9} \pm 0.1$ | $85.0 \pm 0.2$ | $\mathbf{86.5} \pm 0.3$ | $\mathbf{94.9} \pm 0.2$ | $\mathbf{70.5} \pm 0.1$ | **87.0** |
| *Out-of-domain results* | | | | | | |
| ERM | $85.5 \pm 0.2$ | $77.3 \pm 0.4$ | $66.5 \pm 0.3$ | $46.1 \pm 1.8$ | $43.8 \pm 0.1$ | 63.9 |
| CORAL (SOTA) | $86.2 \pm 0.3$ | $78.8 \pm 0.3$ | $68.7 \pm 0.3$ | $47.6 \pm 1.0$ | $41.5 \pm 0.1$ | 64.5 |
| SAM | $85.8 \pm 0.2$ | $79.4 \pm 0.1$ | $69.6 \pm 0.1$ | $43.3 \pm 0.7$ | $44.3 \pm 0.0$ | 64.5 |
| **Domain-Inspired** | $87.3 \pm 0.2$ | $80.1 \pm 0.5$ | $70.7 \pm 0.2$ | $47.9 \pm 0.8$ | $45.8 \pm 0.2$ | 66.4 |
| GSAM | $85.9 \pm 0.1$ | $79.1 \pm 0.2$ | $69.3 \pm 0.0$ | $47.0 \pm 0.8$ | $44.6 \pm 0.2$ | 65.1 |
| **Domain-Inspired** | $87.2 \pm 0.3$ | $80.0 \pm 0.3$ | $70.8 \pm 0.3$ | $\mathbf{50.6} \pm 1.2$ | $45.6 \pm 0.1$ | 66.8 |
| SAGM | $86.6 \pm 0.2$ | $80.0 \pm 0.3$ | $70.1 \pm 0.2$ | $48.8 \pm 0.9$ | $45.0 \pm 0.2$ | 66.1 |
| **Domain-Inspired** | $\mathbf{87.5} \pm 0.3$ | $\mathbf{80.7} \pm 0.2$ | $\mathbf{71.0} \pm 0.2$ | $50.0 \pm 1.2$ | $\mathbf{46.0} \pm 0.1$ | **67.0** |
| + CORAL | $88.4 \pm 0.3$ | $81.2 \pm 0.4$ | $71.7 \pm 0.2$ | $51.7 \pm 0.3$ | $46.3 \pm 0.2$ | 67.9 |

Table 2: **Comparison with state-of-the-art domain generalization methods based on CLIP with ViT-B/16.** Out-of-domain accuracies on five datasets from DomainBed.

| Algorithm | PACS | VLCS | OfficeHome | TerraInc | DomainNet | Avg. |
|---|---|---|---|---|---|---|
| Zero-shot | 96.2 | 81.7 | 82.0 | 33.4 | 57.5 | 70.2 |
| CoOp | 96.8 | 81.2 | 84.2 | 44.9 | 59.9 | 73.4 |
| + SAM | $97.1 \pm 0.1$ | $81.3 \pm 0.8$ | $84.6 \pm 0.2$ | $47.7 \pm 1.3$ | $60.3 \pm 0.2$ | 74.2 |
| + DISAM | $\mathbf{97.2} \pm 0.1$ | $81.8 \pm 0.4$ | $84.8 \pm 0.2$ | $49.5 \pm 1.2$ | $60.6 \pm 0.2$ | 74.8 |
| ERM | $96.1 \pm 0.5$ | $83.0 \pm 0.2$ | $83.3 \pm 0.3$ | $60.9 \pm 0.2$ | $59.9 \pm 0.1$ | 76.7 |
| CLIPOOD[1] | $97.3 \pm 0.1$ | $85.0 \pm 0.4$ | $87.0 \pm 0.2$ | $60.4 \pm 0.7$ | $63.5 \pm 0.1$ | 78.6 |
| CLIPOOD[*2] | $96.6 \pm 0.4$ | $84.1 \pm 0.3$ | $86.1 \pm 0.2$ | $59.7 \pm 0.8$ | $63.1 \pm 0.1$ | 77.9 |
| + SAM | $96.9 \pm 0.2$ | $84.3 \pm 0.6$ | $84.4 \pm 0.4$ | $60.0 \pm 1.4$ | $58.6 \pm 0.2$ | 76.9 |
| + DISAM | $97.1 \pm 0.1$ | $\mathbf{85.6} \pm 0.2$ | $\mathbf{86.6} \pm 0.0$ | $\mathbf{61.1} \pm 0.7$ | $\mathbf{63.6} \pm 0.1$ | **78.8** |

with average improvements of 1.9% for SAM, 1.7% for GSAM, and 1.9% for SAGM. *Notably, SAM performs well when the performance gap between in-domain and out-of-domain is small but worse than ERM on datasets like TerraInc with large gaps, which proves our analysis of SAM's shortcomings under domain shifts.* This shows SAM's inconsistent convergence for large domain shifts, which DISAM addresses by incorporating domain-inspired adaptive adjustments based on domain-level convergence degree. Incorporating CORAL constraints, a recognized effective traditional DG method on DomainBed improves SAGM with DISAM and sets new state-of-the-art results on all settings.

## 4.3 PERFORMANCE UNDER CLIP-BASED PRETRAINED LARGE MODEL

The CLIP-based large pretrained models (Radford et al., 2021) show great zero-shot performance but struggle with domain shifts in downstream tasks. We assess DISAM's out-of-domain results on CLIP using the experimental setup of CLIPOOD (Shu et al., 2023). We test two downstream adaptation methods: CoOp (Zhou et al., 2022a), an effective prompt learning approach, and CLIPOOD, an image encoder finetuning approach for DG problems. For CoOp, we use a 16-length learnable generic prompt and 5000 training steps, and For CLIPOOD settings, we follow Shu et al. (2023). As

---

[1]The original results reported in Shu et al. (2023).

[2]Our reproduced results are based on their official code at https://github.com/thuml/CLIPood.

Table 3: Accuracy on OfficeHome and DomainNet with **both domain shifts and open classes**.

| Split | Algorithm | OfficeHome | | | | DomainNet | | | | | | Avg. |
|---|---|---|---|---|---|---|---|---|---|---|---|---|
| | | A | C | P | R | C | I | P | Q | R | S | |
| **Base** | Zero-shot | 86.7 | 75.9 | 89.6 | 92.2 | 72.6 | 51.8 | 65.4 | 13.6 | 83.5 | 67.2 | 72.6 |
| | CoOp* | 87.3 | 76.7 | 92.2 | 92.5 | 74.6 | 58.2 | 67.9 | 15.0 | 83.7 | 69.9 | 74.4 |
| | +SAM | 89.2 | 79.6 | **93.0** | 93.7 | 73.5 | 58.4 | 67.8 | 14.8 | 83.6 | 69.5 | 75.1 |
| | +DISAM | 88.0 | **80.5** | 92.7 | 92.4 | 75.0 | 59.9 | 68.7 | 14.9 | 84.4 | 70.5 | 75.3 |
| | CLIPOOD* | 88.9 | 79.5 | 92.2 | 94.0 | 76.3 | 58.7 | 69.9 | 17.5 | 85.6 | 72.4 | 76.0 |
| | +SAM | 89.1 | 78.9 | 92.3 | 94.1 | **78.7** | **62.1** | **72.0** | 19.9 | **86.5** | **73.5** | **77.0** |
| | +DISAM | **89.7** | 79.4 | 92.7 | **94.2** | 77.1 | 61.8 | 71.5 | **20.0** | 86.0 | 73.1 | **77.0** |
| **New** | Zero-shot | 76.8 | 59.7 | 88.7 | 86.4 | 69.7 | 45.0 | 67.0 | 14.3 | 83.9 | 60.8 | 67.4 |
| | CoOp* | 73.7 | 56.4 | 86.6 | 85.0 | 69.7 | 47.4 | 67.0 | 15.2 | 82.5 | 61.5 | 66.3 |
| | +SAM | 75.2 | 59.1 | 89.6 | 86.0 | 71.1 | **49.2** | 69.3 | 15.4 | 82.2 | 62.9 | 67.9 |
| | +DISAM | 79.3 | 61.5 | **90.9** | 88.4 | **72.1** | 49.0 | **69.6** | 15.5 | **85.5** | 62.9 | 69.6 |
| | CLIPOOD* | 75.2 | 58.6 | 87.5 | 85.8 | 69.3 | 46.4 | 67.2 | 15.2 | 83.2 | 60.6 | 66.9 |
| | +SAM | 77.2 | 60.0 | 89.8 | 87.6 | 66.8 | 45.4 | 64.9 | 14.8 | 82.0 | 57.1 | 66.9 |
| | +DISAM | **79.7** | **62.0** | 90.5 | **89.0** | 71.8 | 48.7 | 68.7 | **17.5** | 84.7 | **63.0** | **69.7** |
| **Total** | Zero-shot | 82.6 | 67.3 | 88.8 | 89.5 | 71.4 | 47.1 | 66.2 | 13.8 | 83.4 | 63.4 | 69.8 |
| | CoOp* | 81.4 | 65.7 | 88.9 | 88.8 | 71.9 | 51.3 | 67.4 | 15.1 | 83.1 | 65.1 | 70.1 |
| | +SAM | 83.5 | 69.1 | 91.3 | 90.1 | 72.3 | 52.8 | 68.6 | 15.1 | 82.9 | 65.7 | 71.5 |
| | +DISAM | 84.2 | **70.1** | **91.7** | 90.4 | 73.4 | **53.2** | 69.1 | 15.2 | **85.0** | 65.8 | 72.2 |
| | CLIPOOD* | 83.3 | 68.8 | 89.9 | 90.1 | 72.5 | 51.2 | 68.5 | 16.4 | 84.4 | 65.6 | 71.4 |
| | +SAM | 84.2 | 69.2 | 91.0 | 91.0 | 72.3 | 52.0 | 68.4 | 17.3 | 84.3 | 64.0 | 71.8 |
| | + DISAM | **84.6** | 69.5 | 91.3 | **91.2** | **73.5** | **53.2** | **69.4** | **18.3** | 84.9 | **66.3** | **72.6** |

shown in Table 2, DISAM effectively mitigates the impact of domain shifts on model generalization during downstream task adaptation. In addition, as CoOp and CLIPOOD* primarily focus on rapid adaptation with limited parameters, the overfitting risk can be alleviated through early stopping, resulting in the relatively marginal improvements for DISAM in Table 2. Despite this, when handling challenging tasks like TerraInc and DomainNet, our approach still exhibits substantial enhancements.

## 4.4 PERFORMANCE UNDER OPEN-CLASS GENERALIZATION

In this part, we evaluate the performance of our DISAM in a more realistic in-the-wild setting, where both domain shifts and open-class scenarios may arise in the test domain. This setting was first proposed by Shu et al. (2023). OfficeHome and DomainNet are selected to conduct related experiments because they offer an ample number of classes suitable for evaluating open-class situations. To delineate, we segregate the classes within each dataset into two categories, based on the class ID. The initial half denotes the base classes, and the latter half signifies the new classes. Based on Section 4.1, we eliminate the data corresponding to new classes in the training domains. Due to CLIP's open vocabulary property, we can evaluate the new classes on the unseen test domain.

As presented in Table 3, we evaluated the classification accuracy of "Base" classes, "New" classes, and "Total" classes in the test domain, where total classes represent the overall test domain. It revealed that the existing adaptation approach while performing well on base classes, lacks generalization on new classes during the fitting process. Our DISAM mitigates open-class overfitting using domain-level convergence constraints, improving performance by 3.3% over CoOp and 3.1% over CLIPOOD. Figure 4 provides a detailed analysis

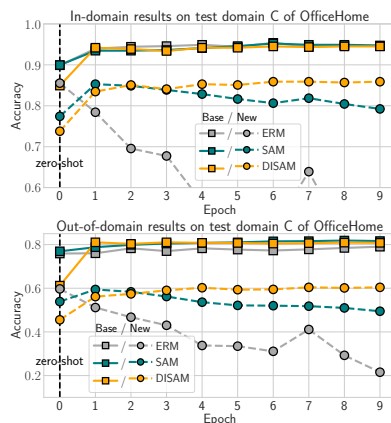

Figure 4: Comparison of CoOp based ERM, SAM and DISAM on accuracy curves for base/new classes. (Top: In-Domain; Bottom: Out-of-Domain)

of open classes and domain shifts dimensions. ERM tends to overfit to both in-domain and base class. While SAM outperforms ERM, it struggles with sharp minima perturbations, failing to effectively escape from them. This difficulty hampers its generalization capabilities in larger models. Please refer to Appendix C.6 for more discussion about DISAM and other methods for open-class generalization.

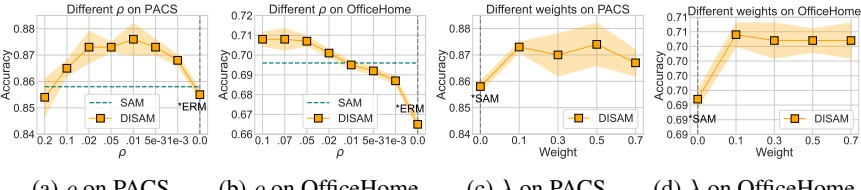

(a) $\rho$ on PACS.  (b) $\rho$ on OfficeHome.  (c) $\lambda$ on PACS.  (d) $\lambda$ on OfficeHome.

Figure 5: Ablation study investigating the sensitivity of hyperparameters, namely **perturbation amplitude** $\rho$ and **variance constraint weight** $\lambda$ in DISAM.

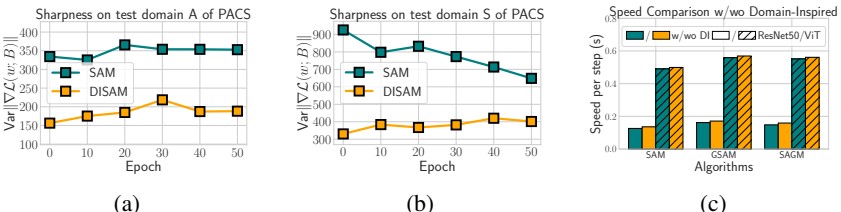

(a)  (b)  (c)

Figure 6: (a) & (b): **Sharpness** curves for SAM and DISAM trained on PACS dataset, which show the trend of the estimated sharpness of the model on the test domain. (c): **Computation cost** with and without Domain-Inspired SAM used on ResNet50 and ViT-B/16 backbone.

## 4.5 ABLATION STUDIES

**Hyperparameter sensitivity.** We performed a sensitivity analysis of the perturbation amplitude $\rho$, and the variance constraint weight $\lambda$, on the PACS and OfficeHome datasets. The default $\rho$ and $\lambda$ is set to 0.05 (Zhuang et al., 2022) and 0.1, respectively. As illustrated in Figure 5(a) and 5(b) within a wide range of $\rho$, DISAM consistently achieves stable and superior results compared to SAM. However, when $\rho$ is too large or small, experimental results worsen. Large $\rho$ hiders convergence, while small $\rho$ weakens sharpness constraint, both affecting generalization. As for $\lambda$, Figure 5(c) and 5(d) show stable results when $\lambda \in [0.1, 0.7]$. However, larger $\lambda$ values increase the variance due to excessive over-conditioning weight, which can also influence the convergence.

**Estimated sharpness on unseen test domain.** Estimating sharpness has a high computational cost. Early methods (Dinh et al., 2017b; Hochreiter & Schmidhuber, 1994b) relied on Monte Carlo sampling, but recent advancements (Jiang et al., 2023; 2020) use gradient-based approximations for efficiency. We assess model sharpness on unseen test domains at each epoch's end, based on the work of Jiang et al. (2023). As depicted in Figure 6(a) and 6(b), our DISAM achieves much smaller gradient variance $\mathrm{Var}\{\nabla\mathcal{L}(w_t; B_t)\}$ than SAM during the whole training, indicating the incorporation of domain-inspired information can further reduce the sharpness of the loss surface.

**Computation cost of DISAM.** In Figure 6(c), we show the extra computational cost from adding **domain-inspired** perturbation direction generation versus the original algorithm (time cost/step, batch size 32, RTX 3090 GPU). Empirical findings show DISAM integration incurs minimal overhead ( 0.01s) across algorithms/backbones, linked solely to domain number and batch size, not model size, via strategic domain loss variance constraints for domain-level convergence consistency.

## 5 CONCLUSION

This paper presents Domain-Inspired Sharpness-Aware Minimization (DISAM), an algorithm that incorporates domain-level convergence consistency into the generation of SAM's perturbations, to address the dilemma under multiple domains. DISAM mitigates SAM's bias in domain shifts that can detrimentally impact the convergence during training, yielding perturbations towards highly converged domains and limiting those in less optimized ones. This is achieved by minimizing the variance of domain loss during perturbation generation, enabling an adaptive weight adjustment for each domain based on its convergence degree, thereby enhancing the convergence across training domains and generalization on unseen domains. Extensive experiments on the domain generalization benchmarks prove DISAM's superiority over existing methods. In addition, DISAM persistents generalization capabilities under parameter-efficient fine-tuning with large models like CLIP.

## ETHICS STATEMENT

This paper does not raise any ethics concerns. This study does not involve any human subjects, practices to data set releases, potentially harmful insights, methodologies and applications, potential conflicts of interest and sponsorship, discrimination/bias/fairness concerns, privacy and security issues, legal compliance, and research integrity issues.

## REPRODUCIBILITY STATEMENT

All experiments were conducted using NVIDIA GeForce RTX 3090 GPU, Python 3.9.15, Pytorch 1.12.1, and clip 1.0. Further details regarding experimental setups and implementation can be found in Section 4.1 and Appendix C, while theoretical proofs are provided in Appendix B. The principal code for implementing Domain-Inspired SAM is presented in Appendix D.

## ACKNOWLEDGMENTS

This work is supported by the National Key R&D Program of China (No. 2022ZD0160702), STCSM (No. 22511106101, No. 18DZ2270700, No. 21DZ1100-100), 111 plan (No. BP0719010), and State Key Laboratory of UHD Video and Audio Production and Presentation. Ruipeng Zhang and Ziqing Fan are partially supported by Wu Wen Jun Honorary Doctoral Scholarship, AI Institute, Shanghai Jiao Tong University.

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

# Appendix

## Table of Contents

# A    RELATED WORK

## A.1    SHARPNESS-AWARE MINIMIZATION (SAM)

Numerous studies (Hochreiter & Schmidhuber, 1994a; Li et al., 2018b; Dinh et al., 2017b) have been conducted to enhance our understanding of the generalization capabilities of deep learning models through an exploration of the geometric properties of the loss landscape. These investigations have consistently demonstrated that deep neural networks exhibiting a flat minimum tend to exhibit superior generalization performance. In order to obtain a flat minimum, the *Sharpness-Aware Minimization* (SAM) approach (Foret et al., 2021) was proposed, which utilizes a base optimizer to simultaneously minimize both the vanilla training loss and the sharpness metric. The sharpness metric, as defined by (Keskar et al., 2017a), quantifies the flatness of a minimum through the eigenvalues of the Hessian matrix. In practice, SAM involves obtaining a fixed-length perturbation through gradient ascent on the initial parameter, followed by updating the gradient based on this perturbed parameter with respect to the initial parameter. Although SAM can result in a flat minimum and substantially enhance the generalization capability, it incurs a twofold increase in computational overhead. The variants of SAM have been extensively investigated from two perspectives: the first pertains to the enhancement of SAM's generalizability (Kwon et al., 2021; Zhuang et al., 2022; Zhang et al., 2023c; Zhao et al., 2022; Wang et al., 2023b), while the second focuses on improving its efficiency (Liu et al., 2022; Du et al., 2022a;b; Mi et al., 2022).

### A.1.1    GENERALIZABILITY IMPROVEMENT OF SAM

One key problem of SAM is that the perturbation obtained by gradient ascent might disagree with sharpness since gradient ascent is only a first-order approximation of the sharpness calculation. Zhuang et al. (2022) introduced a surrogate gap to enhance the evaluation of sharpness, while (Wang et al., 2023b) integrated the perturbed loss and the surrogate gap from (Zhuang et al., 2022) into a unified objective. Additionally, (Zhao et al., 2022) revealed that SAM inherently optimizes both the empirical risk loss and the corresponding gradient norm. Besides, FisherSAM (Kim et al., 2022) and ASAM (Kwon et al., 2021) achieved improved perturbations by adjusting the scales of the perturbation magnitudes. (Zhang et al., 2023c) further proposed Gradient norm Aware Minimization (GAM), which regularized the Hessian of the gradient norm. VaSSO (Li & Giannakis, 2023) focuses on addressing the issue of SAM's subpar performance in perturbation direction generation due to the noise introduced by mini-batch sampling.

### A.1.2    EFFICIENCY IMPROVEMENT OF SAM

Due to the doubled overhead of SAM in comparison to a base optimizer like SGD (Stochastic Gradient Descent), considerable efforts have been devoted to mitigating this overhead. (Liu et al., 2022) introduced LookSAM as a means to reduce the number of perturbations. Meanwhile, (Mi et al., 2022) achieved sparse perturbations through the use of a binary mask. Furthermore, Du *et al.* explored various proxy methods (ESAM (Du et al., 2022a), SAF (Du et al., 2022b)) for computing perturbations, thereby replacing the gradient ascent derivation process employed in SAM.

## A.2    DOMAIN GENERALIZATION

Domain generalization is a vital research direction that focuses on training models capable of generalizing well to unseen domains by leveraging knowledge from multiple source domains (Wang et al., 2023a). Over the past decade, several methods have been proposed to address the challenges of domain generalization. These methods can be broadly categorized into five main approaches: domain alignment, meta-learning, domain hallucination, domain disentanglement, and robustness training. In this section, we provide a brief overview of each of these categories.

### A.2.1    DOMAIN ALIGNMENT-BASED METHOD

The goal of domain alignment is to mitigate discrepancies among distinct source domains by aligning the marginal feature distributions to extract domain-invariant representations. This objective can be accomplished using various strategies, including adversarial training (Li et al., 2018d; Shao et al., 2019), maximum mean discrepancy (Li et al., 2018c), moment matching (Sun & Saenko, 2016),

self-supervised learning (Wang et al., 2020), or contrastive learning (Kim et al., 2021; Motiian et al., 2017; Zhou et al., 2022b; 2023). All of these methods improve generalization across unseen domains by either directly or indirectly reducing the discrepancy between different feature distributions and imposing domain-invariant constraints on these discriminative features.

### A.2.2 META LEARNING-BASED METHODS

These approaches aim to address unforeseen domain shifts and enhance the generalizability of models to such shifts through meta-optimization, achieved by partitioning the training domains into distinct meta-train and meta-test domains. (Li et al., 2018a) first introduced meta learning into DG, following the concept of Modal-Agnostic Meta-Learning (MAML) (Finn et al., 2017). Subsequently, (Balaji et al., 2018) designed a weight regularizer based on the meta-learning framework, while (Li et al., 2019) chose to meta-learn a feature critic loss. (Dou et al., 2019) constrained the invariance of learned semantic relations between the meta-train and meta-test domains. Additionally, (Zhang et al., 2023a) integrated meta learning into a Bayesian framework and enforced the model to learn a meta-variational distribution to enhance knowledge transfer.

### A.2.3 DOMAIN HALLUCINATION-BASED METHODS

Domain hallucination, also known as data augmentation in the presence of domain shifts, aims to encompass a wider range of domain variations by generating additional training samples from fictional domains while preserving their semantic integrity. Early approaches such as (Xu et al., 2021; 2023a; Zhang et al., 2022; Xu et al., 2023b; Zhou et al., 2020a; Yan et al., 2020; Zhou et al., 2020b; Carlucci et al., 2019; Xu et al., 2023c) involve cross-domain data augmentation in the input space and can be categorized into non-parametric and adversarial sample-based approaches. Non-parametric methods (Xu et al., 2021; 2023a; Yan et al., 2020; Zhang et al., 2022) employ traditional image transformations to achieve enhancement, while adversarial sample-based methods (Xu et al., 2023b; Zhou et al., 2020a;b; Carlucci et al., 2019; Xu et al., 2023c) generate samples from a new domain through adversarial training. Adversarial training ensures the quality of generation by enforcing consistency in terms of category among the samples from the generated fictional domain. Some recent work focuses on augmentation in the latent space (Liu et al., 2023; Zhou et al., 2021), which achieves more efficient augmentation perturbations by applying perturbations to the latent features to improve the generalization of the model.

### A.2.4 DOMAIN DISENTANGLEMENT-BASED METHODS

In contrast to the majority of domain generalization approaches that aim for domain invariance, disentanglement-based approaches focus on separating the domain-invariant and domain-specific components. To achieve this, Seo et al. (2020) introduced domain-specific batch normalization (Chang et al., 2019) for each training domain, effectively balancing feature discrimination and invariance. In a similar vein, Jin et al. (2022) designed a style restitution module that encourages the separation of task-relevant and task-irrelevant features. Furthermore, Niu et al. (2023) proposed a two-stage distillation approach, aimed at learning a domain-invariant representation while preserving domain-specific features.

### A.2.5 ROBUSTNESS TRAINING-BASED METHODS

The objective of robustness training-based methods is to incorporate constraints that enhance the model's robustness or flatness during the training process. Robustness-related methods aim to learn domain-invariant representations by employing a technique known as Invariant Risk Minimization (IRM) (Arjovsky et al., 2019). By minimizing the risk across different domains, these methods (Arjovsky et al., 2019; Krueger et al., 2021; Norton & Royset, 2021; Shi et al., 2021; Rame et al., 2022; Li & Giannakis, 2023) seek to learn features that are insensitive to domain variations, thereby improving the model's ability to generalize. On the other hand, a separate class of flatness-related methods (Izmailov et al., 2018; Cha et al., 2021; Zhang et al., 2023b; Foret et al., 2021; Wang et al., 2023b) aims to address the effects of domain shifts by identifying flat minima. These methods strive to find regions in the loss landscape where small perturbations in the input have minimal impact on the model's predictions. By leveraging flat minima, these methods enhance the model's robustness to domain variations.

Table 4: Comparison of SAM-based methods and other robustness training-based methods on the optimization objective.

| Method | Total Optimization Function | Optimization on $w$ | Optimization on $\epsilon$ |
|---|---|---|---|
| ERM | $\min_w \sum_{i=1}^{M} \alpha_i \mathcal{L}_i(w)$ | Same to left | $\times$ |
| V-REx | $\min_w \sum_{i=1}^{M} \alpha_i \mathcal{L}_i(w) + \beta \mathrm{Var}\{\mathcal{L}_i(w)\}_{i=1}^{M}$ | Same to left | $\times$ |
| Fish | $\min_w \sum_{i=1}^{M} \alpha_i \mathcal{L}_i(w) - \gamma \frac{2}{M(M-1)} \sum_{\substack{i,j \in [1,M]}}^{i \neq j} \nabla \mathcal{L}_i(w) \cdot \nabla \mathcal{L}_j(w)$ | Same to left | $\times$ |
| Fishr | $\min_w \sum_{i=1}^{M} \alpha_i \mathcal{L}_i(w) - \lambda \frac{1}{M} \sum_{i=1}^{M} \|\nabla \mathcal{L}_i(w) - \nabla \mathcal{L}(w)\|^2$ | Same to left | $\times$ |
| SAM | $\min_w \max_{\|\epsilon\|_2 \leq \rho} \sum_{i=1}^{M} \alpha_i \mathcal{L}_i(w + \epsilon)$ | $\min_w \sum_{i=1}^{M} \alpha_i \mathcal{L}_i(w + \epsilon)$ | $\max_{\|\epsilon\|_2 \leq \rho} \sum_{i=1}^{M} \alpha_i \mathcal{L}_i(w + \epsilon)$ |
| DISAM | $\min_w \max_{\|\epsilon\|_2 \leq \rho} \left[ \sum_{i=1}^{M} \alpha_i \mathcal{L}_i(w + \epsilon) - \lambda \mathrm{Var}\{\mathcal{L}_i(\hat{w} + \epsilon)\}_{i=1}^{M} \right]$ | $\min_w \sum_{i=1}^{M} \alpha_i \mathcal{L}_i(w + \epsilon)$ | $\max_{\|\epsilon\|_2 \leq \rho} \left[ \sum_{i=1}^{M} \alpha_i \mathcal{L}_i(w + \epsilon) - \lambda \mathrm{Var}\{\mathcal{L}_i(w + \epsilon)\}_{i=1}^{M} \right]$ |

In Table 4, we provide the comparison of optimization objectives between representative algorithms in the two categories. Domain-invariant methods solely concentrate on optimizing the parameter $w$. For instance, V-REx (Krueger et al., 2021) directly minimizes the variance of the domain loss, which can have a detrimental effect on convergence. Similarly, Fish (Shi et al., 2021) and Fishr (Rame et al., 2022) impose constraints on gradient updates. SAM-based methods require the estimation of sharpness, so in addition to optimizing the parameters $w$, they also need to optimize the perturbation directions $\epsilon$.

This paper primarily concentrates on the flatness-based method, which encompasses two main approaches for enhancing the flatness of the model. The first approach involves leveraging the self-ensemble of multiple minima attained during the training process to passively acquire a result that favors flatness minima. Notable examples of this approach include Stochastic Weight Averaging (SWA) (Izmailov et al., 2018) and Stochastic Weight Averaging Densely (SWAD) (Cha et al., 2021). The second approach involves directly optimizing for flatness and is referred to as Sharpness-Aware Minimization (SAM) (Foret et al., 2021). In the subsequent section, we will delve into a comprehensive review of the relevant literature pertaining to these approaches.

## B  DETAILS OF DISAM

### B.1  COMPARATIVE ANALYSIS OF DISAM VERSUS GENERAL CONVERGENCE CONSISTENCY

Here, we present a thorough examination of the distinctions between our proposed DISAM framework and the broader, conventional convergence consistency issue like V-REx(Krueger et al., 2021) and Fishr(Rame et al., 2022). Specifically, we address the following aspects:

- **Distinct Focus:** DISAM focuses on the issue where SAM-based methods are unable to accurately estimate sharpness in domain shift scenarios, leading to the ineffective sharpness minimization and reduction in generalization performance.
- **Enhancing on Top of General Methods:** While traditional solutions(Krueger et al., 2021; Rame et al., 2022; Shi et al., 2021) aim at convergence consistency in parameter optimization, DISAM's methodology is distinct and orthogonal. It builds upon methods like V-REx(Krueger et al., 2021) and Fishr(Rame et al., 2022), but goes further in enhancing out-of-domain generalization through better sharpness minimization. This is evident in our experiments, where combining DISAM with Fishr results in significant performance gains (shown in Table 5).

We also provide extensive experimental results to validate DISAM's effectiveness and its practical implications in various domain-shift scenarios.

It is imperative to reiterate the contributions of our DISAM. We provide a detailed exposition of how simplistic applications of SAM compromise training robustness, especially when dealing with domain shifts. DISAM strategically mitigates these issues by finely tuning the perturbation vectors and their

Table 5: Comparison with other general convergence consistency methods.

| Algorithm | PACS | VLCS | OfficeHome | TerraInc | DomainNet | Avg. |
|---|---|---|---|---|---|---|
| V-REx | 84.9 | 78.3 | 66.4 | 46.4 | 33.6 | 61.9 |
| **V-REx + DISAM** | 85.8 | 78.4 | 70.5 | 45.9 | 42.3 | 64.6 |
| Fishr | 86.9 | 78.2 | 68.2 | 53.6 | 41.8 | 65.7 |
| **Fishr + DISAM** | 87.5 | 79.2 | 70.7 | 54.8 | 43.9 | **67.2** |

---

**Algorithm 1** Domain-Inspired Sharpness-Aware Minimization (DISAM).

---

**Input:** Source Domains $\mathcal{S} = \{D_1, \cdots, D_M\}$, initial model $w_1$, perturbation ratio $\rho$, variance constraint weight $\lambda$, learning rate $\eta_t$, training iterations $T$.
**Output:** Generalization model $w_T$.

1: **for** $t$ in $1 \cdots T$ **do**
2:      Sample mini-batch $B = \{B_1, \cdots, B_M\} \subseteq \mathcal{S}$, where $B_i \subseteq D_i$ and $|B_i| \geq 0$.
3:      Compute the domain-inspired loss gradient:
       $\nabla \mathcal{L}_{DI}(w_t; B) = \nabla \mathcal{L}(w_t; B) - \lambda \nabla \text{Var}\{\mathcal{L}_i(w_t); B_i\}_{i=1}^M$.
4:      Get the perturbation weight: $w_t^{asc} = w_t + \rho \frac{\nabla \mathcal{L}_{DI}(w_t; B)}{\|\nabla \mathcal{L}_{DI}(w_t; B)\|}$.
5:      Update weights: $w_{t+1} = w_t - \eta_t \nabla \mathcal{L}(w_t^{asc}; B)\}$.
6: **end for**

---

location points, thus significantly enhancing model generalization. Furthermore, we underscore the notable enhancements achieved with DISAM, as corroborated by comprehensive experimental analyses and the ensuing performance metrics.

### B.2   ALGORITHM OF DISAM

We give specific algorithmic details for DISAM in Algorithm 1, and the python code implementation is in Appendix D.

### B.3   PROOF OF DISAM'S CONVERGENCE

**Theorem 1.** *(Convergence During Training). Consider a non-convex function $\mathcal{L}(w)$ with Lipschitz-smooth constant $L$ and lower bound $\mathcal{L}_{min}$. With the bounded norm assumption of noisy stochastic gradients ($\|\nabla \mathcal{L}_p(w)\|_2 \leq G$) at the t-step, the learning rate $\eta_t = \eta_0/\sqrt{t}$ and a fixed perturbation amplitude $\rho$, we have:*

$$\frac{1}{T} \sum_{t=1}^{T} \mathbb{E}\|\nabla \mathcal{L}_p(w_t)\|_2^2 \leq \frac{\mathcal{L}_p(w_0) - \mathcal{L}_{min}}{\eta_0} \frac{1}{\sqrt{T}} + \frac{(LG^2 + \rho^2 L\Gamma^2)\eta_0 \log(T)}{\sqrt{T}} \quad (9)$$

*where in SAM, $\Gamma = L$ and when use DISAM $\Gamma \leq L$.*

*Proof.* For simplicity of notation, we denote the update at step $t$ as $d_t = -\eta_t g_p^{(t)}$, where $\eta_t$ is the decayed learning rate and $g_p^t$ is the expected gradient of perturbation loss $\mathcal{L}_p$. By $L$-smoothness of the loss function $\mathcal{L}$ and the definition of $\mathcal{L}_p(w_t) = \mathcal{L}(w_t^{asc})$, where $w_t^{asc}$ represents the parameters after the perturbation of gradient ascent, we have:

$$\mathcal{L}_p(w_{t+1}) = \mathcal{L}(w_{t+1}^{asc}) \leq \mathcal{L}(w_t^{asc}) + \langle \nabla \mathcal{L}(w_t^{asc}), w_{t+1}^{asc} - w_t^{asc} \rangle + \frac{L}{2}\|w_{t+1}^{asc} - w_t^{asc}\|^2 \quad (10)$$

where $L$ is the Lipschitz constant of loss $\mathcal{L}$ and with the definition of $d_t = w_{t+1} - w_t$ and $w_t^{asc} = w_t + \epsilon_t$, we have:

$$\mathcal{L}_p(w_{t+1}) \leq \mathcal{L}(w_t^{asc}) + \langle \nabla \mathcal{L}(w_t^{asc}), w_{t+1} + \epsilon_{t+1} - w_t - \epsilon_t \rangle + \frac{L}{2}\|w_{t+1} + \epsilon_{t+1} - w_t - \epsilon_t\|^2$$
$$\leq \mathcal{L}(w_t^{asc}) + \langle \nabla \mathcal{L}(w_t^{asc}), d_t \rangle + L\|d_t\|^2 + \langle \nabla \mathcal{L}(w_t^{asc}), \epsilon_{t+1} - \epsilon_t \rangle + L\|\epsilon_{t+1} - \epsilon_t\|^2 \quad (11)$$

Let us take the expectation conditioned on observations up to step $t$. For the sake of simplicity, we use the symbol $\mathbb{E}$ to denote the expectation over all possible data points on the training data distribution. Moreover, given the observations up to step $t$, we can use the definition of $d_t$ to obtain:

$$
\begin{aligned}
\mathbb{E}[\mathcal{L}_p(w_{t+1})] &\leq \mathcal{L}(w_t^{asc}) - \eta_t \langle \nabla \mathcal{L}(w_t^{asc}), \mathbb{E}[g_p^{(t)}] \rangle + \eta_t^2 L \mathbb{E} \| g_p^{(t)} \|^2 \\
&\quad + \mathbb{E} \langle \nabla \mathcal{L}(w_t^{asc}), \epsilon_{t+1} - \epsilon_t \rangle + L \mathbb{E} \| \epsilon_{t+1} - \epsilon_t \|^2 \\
&\leq \mathcal{L}(w_t^{asc}) - \eta_t \mathbb{E} \| \nabla \mathcal{L}(w_t^{asc}) \|_2^2 + \eta_t^2 L G^2 \\
&\quad + \mathbb{E} \langle \nabla \mathcal{L}(w_t^{asc}), \epsilon_{t+1} - \epsilon_t \rangle + L \mathbb{E} \| \epsilon_{t+1} - \epsilon_t \|^2
\end{aligned}
\tag{12}
$$

By the definition of $\epsilon_t$, we have:

$$
\epsilon_t = \rho \frac{g^{(t)}}{\|g^{(t)}\|}, \epsilon_{t+1} = \rho \frac{g^{(t+1)}}{\|g^{(t+1)}\|}
\tag{13}
$$

where $g^{(t)}$ is the gradient of $\mathcal{L}$ at $w_t$ with the domain-inspired gradient in Eq.( 8). We denote $\epsilon_t = \nabla \mathcal{L}_d(w_t) = \sum_{i=1}^M \beta_t^i \nabla \mathcal{L}^i(w_t)$, where $\beta_t^i = \alpha_i - \frac{2\lambda}{M} \left( \mathcal{L}^i(w_t) - \frac{1}{M} \sum_{j=1}^M \mathcal{L}^j(w_t) \right)$. Since both $\epsilon_t$ and $\epsilon_{t+1}$ are unit length vectors, $\epsilon_{t+1} - \epsilon_t$ can be bounded by the arc length $\phi_t$ between them. Here the difference vector between $\epsilon_{t+1}$ and $\epsilon_t$ can be regarded as a random noise in the gradient direction and in SAM $\rho \gg \eta_t$, so the expectation of the inner product with the gradient direction $\nabla \mathcal{L}(w_t^{asc})$ can be approximated as 0 ($w_t^{asc}$ is updated from $w_t$ with a larger step size $\rho$, and its gradient direction can be considered approximately independent of the gradient direction in the neighborhood of $w_t$, so its difference with the inner product between $\epsilon_{t+1}$ and $\epsilon_t$ is negligible). Therefore, we have:

$$
\begin{aligned}
\mathbb{E}[\mathcal{L}_p(w_{t+1})] &\leq \mathcal{L}(w_t^{asc}) - \eta_t \mathbb{E} \| \nabla \mathcal{L}(w_t^{asc}) \|_2^2 + \eta_t^2 L G^2 \\
&\quad + \mathbb{E}[\langle \nabla \mathcal{L}(w_t^{asc}), \epsilon_{t+1} \rangle - \langle \nabla \mathcal{L}(w_t^{asc}), \epsilon_t \rangle] + L \rho^2 \mathbb{E} \left\| \frac{g^{(t+1)}}{\|g^{(t+1)}\|} - \frac{g^{(t)}}{\|g^{(t)}\|} \right\|^2 \\
&\leq \mathcal{L}(w_t^{asc}) - \eta_t \mathbb{E} \| \nabla \mathcal{L}(w_t^{asc}) \|_2^2 + \eta_t^2 L G^2 + L \rho^2 \phi_t^2
\end{aligned}
\tag{14}
$$

Because of the continuity of the optimization, the angle between the gradient perturbations before and after is small. Therefore, we can get:

$$
\begin{aligned}
\phi_t &\approx \tan \phi_t = \frac{\|\epsilon_{t+1} - \epsilon_t\|}{\|\epsilon_t\|} + O(\phi_t^2) = \frac{\|\nabla \mathcal{L}_d(w_{t+1}) - \nabla \mathcal{L}_d(w_t)\|}{\|\nabla \mathcal{L}_d(w_t)\|} + O(\phi_t^2) \\
&= \frac{\| \sum_{i=1}^M \left( \beta_{t+1}^i \nabla \mathcal{L}^i(w_{t+1}) - \beta_t^i \nabla \mathcal{L}^i(w_t) \right) \|}{\|\nabla \mathcal{L}_d(w_t)\|} + O(\phi_t^2) \\
&= \frac{\| \sum_{i=1}^M \left( (\beta_{t+1}^i - \beta_t^i) \nabla \mathcal{L}^i(w_{t+1}) + \beta_t^i (\nabla \mathcal{L}^i(w_{t+1}) - \nabla \mathcal{L}^i(w_t)) \right) \|}{\|\nabla \mathcal{L}_d(w_t)\|} + O(\phi_t^2)
\end{aligned}
\tag{15}
$$

Here we consider the effect of the weight coefficients generated by DISAM in the perturbation of $\nabla \mathcal{L}_d$, for the part of $\mathcal{L}^i(w_t)$ that is large, $\beta_t^i$ is smaller, we assume that the larger $\mathcal{L}^i(w_t)$ is, the larger the corresponding gradient $\nabla \mathcal{L}^i(w_t)$ is also, and after one optimization process, the variability between the domains will be reduced, so $\beta_{t+1}^i$ is a little bit smaller than the weight of $\beta_t^i$, in the place where the gradient is large, and by the rearranging inequality, we can obtained:

$$
\sum_{i=1}^M \beta_{t+1}^i \nabla \mathcal{L}^i(w_{t+1}) \leq \sum_{i=1}^M \beta_{t+1}^i \nabla \mathcal{L}^i(w_t)
\tag{16}
$$

So bring Eq.( 16) to Eq.( 15), and with $\nabla \mathcal{L}(w_{t+1}) = \nabla \mathcal{L}(w_t + d_t) = \nabla \mathcal{L}(w_t) + H d_t + O(\|d_t\|^2)$ we can get:

$$
\phi_t \leq \frac{\| \sum_{i=1}^M (\beta_{t+1}^i - \beta_t^i) \nabla \mathcal{L}^i(w_{t+1}) + H d_t + O(\|d_t\|^2) \|}{\|\nabla \mathcal{L}_d(w_t)\|} + O(\phi_t^2) \leq \eta_t \Gamma
\tag{17}
$$

Here since $\sum_{i=1}^{M}(\beta_{t+1}^i - \beta_t^i)\nabla\mathcal{L}^i(w_{t+1}) \leq 0$, we use $\Gamma$ to denote an upper bound that is smaller than $L$.

Plug Eq.( 17) into Eq.( 14), we have:

$$\mathbb{E}[\mathcal{L}_p(w_{t+1})] \leq \mathcal{L}(w_t^{asc}) - \eta_t\mathbb{E}\|\nabla\mathcal{L}(w_t^{asc})\|_2^2 + \eta_t^2 LG^2 + L\rho^2\eta_t^2\Gamma^2 \tag{18}$$

Perform telescope sum and note that $\eta_T = \frac{\eta_0}{\sqrt{T}}$, we have:

$$
\begin{aligned}
\mathbb{E}\mathcal{L}_p(w_T) - \mathcal{L}_p(w_0) &\leq -\sum_{t=1}^{T}\eta_t\mathbb{E}\|\nabla\mathcal{L}(w_t^{asc})\|_2^2 + (LG^2 + \rho^2 L\Gamma^2)\eta_0^2\sum_{t=1}^{T}\frac{1}{t} \\
&\leq -\sum_{t=1}^{T}\eta_t\mathbb{E}\|\nabla\mathcal{L}(w_t^{asc})\|_2^2 + (LG^2 + \rho^2 L\Gamma^2)\eta_0^2\log(T)
\end{aligned}
\tag{19}
$$

Hence,

$$\eta_T\sum_{t=1}^{T}\mathbb{E}\|\nabla\mathcal{L}(w_t^{asc})\|_2^2 \leq \sum_{t=1}^{T}\eta_t\mathbb{E}\|\nabla\mathcal{L}(w_t^{asc})\|_2^2 \leq \mathcal{L}_p(w_0) - \mathcal{L}_m in + (LG^2 + \rho^2 L\Gamma^2)\eta_0^2\log(T) \tag{20}$$

Note that $\eta_T = \frac{\eta_0}{\sqrt{T}}$, we have:

$$\frac{1}{T}\sum_{t=1}^{T}\mathbb{E}\|\nabla\mathcal{L}(w_t^{asc})\|_2^2 \leq \frac{\mathcal{L}_p(w_0) - \mathcal{L}_m in}{\eta_0}\frac{1}{\sqrt{T}} + \frac{(LG^2 + \rho^2 L\Gamma^2)\eta_0\log(T)}{\sqrt{T}} \tag{21}$$

$\square$

**The influence of $\lambda$:** In the proof of Theorem 1, specifically in Eq. (15), $\lambda$ is integrated into $\beta$, serving as a hyperparameter that regulates the weight adjustment in DISAM. It functions by modulating the degree of correction for domain shifts:

$$\beta_t^i = \alpha_i - \frac{2\lambda}{M}\left(\mathcal{L}^i(w_t) - \frac{1}{M}\sum_{j=1}^{M}\mathcal{L}^j(w_t)\right)$$

The choice of $\lambda$ influences how aggressively DISAM responds to variance or domain shifts, with a higher $\lambda$ leading to more pronounced adjustments in $\beta$. Our experimental analysis in Figure 5(c) and 5(d), reveals that DISAM's performance remains relatively stable across a wide range of $\lambda$ values. However, choosing too large $\lambda$ can result in overly aggressive early training adjustments, yielding the increased variance in repeated experiments. Consequently, we adopted a default $\lambda$ value of 0.1 in all experiments.

### B.4 DISCUSSION OF THE ROLE OF $\rho$ IN DISAM

Here, we provide a detailed discussion on how $\rho$ affects both generalization and convergence. First, we introduce the generalization theorem of the upper bound on generalization error, which is only related to the magnitude of $\rho$, and DISAM follows the same upper bound on generalization error as SAM. In the SAM framework, the parameter $\rho$ plays a crucial role in determining generalizability. As established in SAM (Foret et al., 2021), there exists an upper bound on the generalization error for SAM, suggesting that a larger $\rho$ could potentially enhance generalization, provided that convergence is not impeded. Here is the relevant generalization bound from SAM (Foret et al., 2021):

**Theorem 2.** *(Generalization Bound of SAM). For any $\rho > 0$ and any distribution $\mathcal{D}$, with probability $1 - \delta$ over the choice of the training set $S \sim \mathcal{D}$,*

$$\mathcal{L}_{\mathcal{D}}(w) \leq \max_{\|\epsilon\|_2 \leq \rho}\mathcal{L}_S(w + \epsilon) + \sqrt{\frac{k\log\left(1 + \frac{\|w\|_2^2}{\rho^2}(1 + \sqrt{\frac{\log(n)}{k}})^2\right) + 4\log\frac{n}{\delta} + \tilde{O}(1)}{n - 1}} \tag{22}$$

*where $n = |S|$, $k$ is the number of parameters and we assumed $\mathcal{L}_{\mathcal{D}}(w) \leq \mathbb{E}_{\epsilon_i \approx \mathcal{N}(0,\rho)}[\mathcal{L}_{\mathcal{D}}(w + \epsilon)]$. This theorem's proof focuses solely on the magnitude of $\rho$, thus affirming the applicability of this theoretical framework to DISAM.*

When considering domain shift, the upper bound on generalization error for the test domain is:

**Theorem 3.** *(**PAC-Bayesian Generalization Bound**). For any $\rho > 0$ and the unseen domain $T$, suppose we have multi-source domains $S = \{S_1, S_2, \cdots\}$ with a total of $N$ samples. Let $\mathcal{H}$ be the hypothesis space and $\Omega$ be the corresponding parameter space, where the VC dimension of $\mathcal{H}$ is $d$. We denote the domain divergence between two domains $D_i$ and $D_j$ on the hypothesis space $\mathcal{H}$ as $d_{\mathcal{H}\Delta\mathcal{H}}(D_i, D_j)$. Then, for any $\delta \in (0, 1)$, with probability at least $1 - \delta$, for all $w \in \Omega$, we have:*

$$\mathcal{L}_T(w) \leq \max_{\|\epsilon\|_2 \leq \rho} \mathcal{L}_S(w + \epsilon) + \frac{1}{2} d_{\mathcal{H}\Delta\mathcal{H}}(S, T) + \sqrt{\frac{\log d + \log \frac{1}{\delta}}{2N}} + \lambda$$
$$+ \sqrt{\frac{k \log \left(1 + \frac{\|w\|_2^2}{\rho^2} \left(1 + \sqrt{\frac{\log(N)}{k}}\right)^2 + 4 \log \frac{N}{\delta} + \tilde{O}(1)\right)}{N - 1}} \tag{23}$$

*where $\lambda$ is the optimal combined risk on $T$ and $S$ that can be achieved by the parameters in $\Omega$.*

Combining this with the convergence theorem (Theorem 1), there is a trade-off with respect to $\rho$. A larger $\rho$ might theoretically enhance generalization but poses greater challenges for convergence. This reflects the intuitive notion that searching for flatter minima across a broader range is inherently more complex, which can potentially affect training efficiency. However, if $\mathcal{L}_S(w + \epsilon)$ can be converged with a sufficiently small value, a larger $\rho$ corresponds to better generalization. DISAM, compared to SAM, converges faster, which means that under the same convergence speed, a larger $\rho$ can be used to achieve better generalization. This advantage is empirically showcased in Figure 3(c) and (d), where we demonstrate that DISAM effectively employs a larger $\rho$ compared to traditional SAM. This ensures both convergence and enhanced generalization. Such a capability to balance between convergence efficiency and generalization is a distinguishing feature of DISAM over conventional SAM methods.

## C DETAILED EXPERIMENTS

### C.1 DETAILED EXPERIMENT SETUPS

We present the detailed results obtained from five datasets, namely PACS (Li et al., 2017) (9,991 images, 7 classes, 4 domains), VLCS (Fang et al., 2013) (10,729 images, 5 classes, 4 domains), OfficeHome (Venkateswara et al., 2017) (15,588 images, 65 classes, 4 domains), TerraIncognita (Beery et al., 2018) (abbreviated as TerraInc, 24,788 images, 10 classes, 4 domains), and DomainNet (Peng et al., 2019) (586,575 images, 345 classes, 6 domains), following the DomainBed benchmark (Gulrajani & Lopez-Paz, 2021) with the ResNet50 backbone architecture. We set the hyperparameters for the Domain-Inspired + SAM method as follows: $\rho = 0.5$ and $\lambda = 0.1$ for PACS, VLCS, OfficeHome, and DomainNet; for TerraInc, we use $\rho = 0.01$ and $\lambda = 0.2$. Both Domain-Inspired + GSAM and Domain-Inspired + SAGM employ the strategy described in the supplementary material of SAGM (Wang et al., 2023b). As for the CoOp with CLIP, we set the batch size as 32 and the default learning rate as 2e-3. Given the detailed experimental hyperparameter settings provided in the SAGM supplement (Wang et al., 2023b) and the official open-source CLIPOOD code (Shu et al., 2023), we directly applied these official settings. The results, replicated using the official open-source CLIPOOD code, are presented in Table 2 of the main text.

As for the experiments on open class, we found that CLIPOOD requires a lower learning rate and correspondingly lower $\rho$, and therefore used learning rate 1e-07 and $\rho$ 1e-05 as default settings.

### C.2 DETAILED EXPERIMENTAL RESULTS

We present the specific out-of-domain experimental results for each dataset in Table 1, corresponding to each leave-one-domain-out setting.

Table 6: **Comparison with state-of-the-art domain generalization methods.** Out-of-domain accuracies on the PACS dataset with ResNet50 backbone.

| Algorithm | Art | Cartoon | Photo | Sketch | Avg. |
|---|---|---|---|---|---|
| ERM | $84.7_{\pm0.4}$ | $80.0_{\pm0.6}$ | $97.2_{\pm0.3}$ | $79.3_{\pm1.0}$ | 85.5 |
| SAM | $85.6_{\pm2.1}$ | $80.9_{\pm1.2}$ | $97.0_{\pm0.4}$ | $79.6_{\pm1.6}$ | 85.8 |
| **Domain-Inspired** | $87.1_{\pm0.4}$ | $81.9_{\pm0.5}$ | $96.2_{\pm0.3}$ | $\mathbf{83.1}_{\pm0.7}$ | 87.1 |
| GSAM | $86.9_{\pm0.1}$ | $80.4_{\pm0.2}$ | $97.5_{\pm0.0}$ | $78.7_{\pm0.8}$ | 85.9 |
| **Domain-Inspired** | $88.4_{\pm0.2}$ | $81.1_{\pm0.3}$ | $97.0_{\pm0.0}$ | $82.3_{\pm0.6}$ | 87.2 |
| SAGM | $87.4_{\pm0.2}$ | $80.2_{\pm0.3}$ | $\mathbf{98.0}_{\pm0.2}$ | $80.8_{\pm0.6}$ | 86.6 |
| **Domain-Inspired** | $\mathbf{89.7}_{\pm0.6}$ | $\mathbf{81.5}_{\pm0.0}$ | $97.0_{\pm0.1}$ | $81.8_{\pm0.6}$ | **87.5** |
| +CORAL | $89.8_{\pm0.5}$ | $82.9_{\pm0.2}$ | $97.4_{\pm0.2}$ | $83.4_{\pm0.2}$ | **88.4** |

Table 7: **Comparison with state-of-the-art domain generalization methods.** Out-of-domain accuracies on the VLCS dataset with ResNet50 backbone.

| Algorithm | Caltech | LabelMe | Pascal | Sun | Avg. |
|---|---|---|---|---|---|
| ERM | $98.0_{\pm0.3}$ | $64.7_{\pm1.2}$ | $71.4_{\pm1.2}$ | $75.2_{\pm1.6}$ | 77.3 |
| SAM | $99.1_{\pm0.2}$ | $65.0_{\pm1.0}$ | $73.7_{\pm1.0}$ | $79.8_{\pm0.1}$ | 79.4 |
| **Domain-Inspired** | $99.3_{\pm0.0}$ | $66.3_{\pm0.5}$ | $81.0_{\pm0.1}$ | $73.2_{\pm0.1}$ | 79.9 |
| GSAM | $98.7_{\pm0.3}$ | $64.9_{\pm0.2}$ | $74.3_{\pm0.0}$ | $78.5_{\pm0.8}$ | 79.1 |
| **Domain-Inspired** | $99.8_{\pm0.0}$ | $\mathbf{66.6}_{\pm0.1}$ | $74.2_{\pm0.9}$ | $79.3_{\pm0.1}$ | 80.0 |
| SAGM | $99.0_{\pm0.2}$ | $65.2_{\pm0.4}$ | $\mathbf{75.1}_{\pm0.3}$ | $80.7_{\pm0.8}$ | 80.0 |
| **Domain-Inspired** | $\mathbf{99.9}_{\pm0.1}$ | $66.1_{\pm0.6}$ | $\mathbf{75.1}_{\pm0.3}$ | $\mathbf{81.8}_{\pm0.0}$ | **80.7** |
| +CORAL | $99.7_{\pm0.1}$ | $67.8_{\pm0.7}$ | $75.5_{\pm0.8}$ | $81.6_{\pm0.2}$ | **81.2** |

Table 8: **Comparison with state-of-the-art domain generalization methods.** Out-of-domain accuracies on the OfficeHome dataset with ResNet50 backbone.

| Algorithm | Art | Clipart | Product | Real World | Avg. |
|---|---|---|---|---|---|
| ERM | $61.3_{\pm0.7}$ | $52.4_{\pm0.3}$ | $75.8_{\pm0.1}$ | $76.6_{\pm0.3}$ | 66.5 |
| SAM | $64.5_{\pm0.3}$ | $56.5_{\pm0.2}$ | $77.4_{\pm0.1}$ | $79.8_{\pm0.4}$ | 69.6 |
| **Domain-Inspired** | $65.8_{\pm0.2}$ | $55.6_{\pm0.2}$ | $79.2_{\pm0.2}$ | $80.6_{\pm0.1}$ | 70.3 |
| GSAM | $64.9_{\pm0.1}$ | $55.2_{\pm0.2}$ | $77.8_{\pm0.0}$ | $79.2_{\pm0.2}$ | 69.3 |
| **Domain-Inspired** | $65.7_{\pm0.3}$ | $\mathbf{57.4}_{\pm0.3}$ | $79.4_{\pm0.1}$ | $80.7_{\pm0.3}$ | 70.8 |
| SAGM | $65.4_{\pm0.4}$ | $57.0_{\pm0.3}$ | $78.0_{\pm0.3}$ | $80.0_{\pm0.2}$ | 70.1 |
| **Domain-Inspired** | $\mathbf{67.2}_{\pm0.0}$ | $56.3_{\pm0.3}$ | $\mathbf{79.6}_{\pm0.2}$ | $\mathbf{81.0}_{\pm0.3}$ | **71.0** |
| +CORAL | $68.5_{\pm0.1}$ | $57.6_{\pm0.1}$ | $79.3_{\pm0.4}$ | $81.3_{\pm0.2}$ | **71.7** |

Table 9: **Comparison with state-of-the-art domain generalization methods.** Out-of-domain accuracies on the TerraInc dataset with ResNet50 backbone.

| Algorithm | L100 | L38 | L43 | L46 | Avg. |
|---|---|---|---|---|---|
| ERM | $49.8_{\pm4.4}$ | $42.1_{\pm1.4}$ | $56.9_{\pm1.8}$ | $35.7_{\pm3.9}$ | 46.1 |
| SAM | $46.3_{\pm1.0}$ | $38.4_{\pm2.4}$ | $54.0_{\pm1.0}$ | $34.5_{\pm0.8}$ | 43.3 |
| **Domain-Inspired** | $46.2_{\pm2.9}$ | $41.6_{\pm0.1}$ | $58.0_{\pm0.5}$ | $40.5_{\pm2.2}$ | 46.6 |
| GSAM | $50.8_{\pm0.1}$ | $39.3_{\pm0.2}$ | $\mathbf{59.6}_{\pm0.0}$ | $38.2_{\pm0.8}$ | 47.0 |
| **Domain-Inspired** | $56.7_{\pm1.5}$ | $\mathbf{46.7}_{\pm1.0}$ | $59.2_{\pm0.7}$ | $39.9_{\pm1.5}$ | **50.6** |
| SAGM | $54.8_{\pm1.3}$ | $41.4_{\pm0.8}$ | $57.7_{\pm0.6}$ | $\mathbf{41.3}_{\pm0.4}$ | 48.8 |
| **Domain-Inspired** | $\mathbf{57.6}_{\pm1.6}$ | $44.8_{\pm1.5}$ | $58.6_{\pm1.2}$ | $38.9_{\pm0.6}$ | 50.0 |
| + CORAL | $57.9_{\pm0.3}$ | $46.6_{\pm0.6}$ | $59.9_{\pm0.3}$ | $42.5_{\pm0.1}$ | 51.7 |

Table 10: **Comparison with state-of-the-art domain generalization methods.** Out-of-domain accuracies on the DomainNet dataset with ResNet50 backbone.

| Algorithm | Clipart | Infograph | Painting | Quickdraw | Real | Sketch | Avg. |
|---|---|---|---|---|---|---|---|
| ERM | $62.8_{\pm0.4}$ | $20.2_{\pm0.3}$ | $50.3_{\pm0.3}$ | $13.7_{\pm0.5}$ | $63.7_{\pm0.2}$ | $52.1_{\pm0.5}$ | 43.8 |
| SAM | $64.5_{\pm0.3}$ | $20.7_{\pm0.2}$ | $50.2_{\pm0.1}$ | $15.1_{\pm0.3}$ | $62.6_{\pm0.2}$ | $52.7_{\pm0.3}$ | 44.3 |
| **Domain-Inspired** | $\mathbf{65.9}_{\pm0.2}$ | $20.7_{\pm0.2}$ | $51.7_{\pm0.3}$ | $\mathbf{16.6}_{\pm0.3}$ | $62.8_{\pm0.5}$ | $\mathbf{54.8}_{\pm0.4}$ | 45.4 |
| GSAM | $64.2_{\pm0.3}$ | $20.8_{\pm0.2}$ | $50.9_{\pm0.0}$ | $14.4_{\pm0.8}$ | $63.5_{\pm0.2}$ | $53.9_{\pm0.2}$ | 44.6 |
| **Domain-Inspired** | $65.7_{\pm0.1}$ | $21.3_{\pm0.1}$ | $52.2_{\pm0.1}$ | $15.6_{\pm0.0}$ | $64.5_{\pm0.2}$ | $54.1_{\pm0.2}$ | 45.6 |
| SAGM | $64.9_{\pm0.2}$ | $21.1_{\pm0.3}$ | $51.5_{\pm0.2}$ | $14.8_{\pm0.2}$ | $64.1_{\pm0.2}$ | $53.6_{\pm0.2}$ | 45.0 |
| **Domain-Inspired** | $\mathbf{65.9}_{\pm0.2}$ | $\mathbf{21.4}_{\pm0.0}$ | $\mathbf{52.6}_{\pm0.1}$ | $15.8_{\pm0.0}$ | $\mathbf{65.3}_{\pm0.0}$ | $\mathbf{54.8}_{\pm0.2}$ | **46.0** |
| +CORAL | $66.4_{\pm0.3}$ | $21.9_{\pm0.2}$ | $53.1_{\pm0.1}$ | $16.1_{\pm0.0}$ | $65.3_{\pm0.0}$ | $55.0_{\pm0.0}$ | 46.3 |

## C.3 DETAILS ABOUT ESTIMATED SHARPNESS ON UNSEEN TEST DOMAIN

Estimating sharpness involves a significant computational overhead. In the earliest methods, Monte Carlo random sampling was the only viable approach (Dinh et al., 2017b; Hochreiter & Schmidhuber, 1994b). However, recent advancements have introduced efficient approximation techniques based on gradients to estimate sharpness (Jiang et al., 2023; 2020). Based on the work of Jiang et al. (2023), we assess the sharpness of the training model on the unseen test domain at the end of each epoch. Sharpness is commonly characterized by the eigenvalues of the Hessian matrix (Keskar et al., 2017b; Dinh et al., 2017a), but direct computation incurs substantial overhead. To address this, a computationally efficient measurement of sharpness is proposed by Jiang et al. (2020), which utilizes the gradient variance $\text{Var}\{\nabla\mathcal{L}(w_t; B_t)\}$ as an estimate ($B_t$ represent the batch data sampled at step $t$).

## C.4 DETAILS ABOUT COMPARISON OF COMPUTATION COST

We selected the PACS dataset for experimentation, using a platform with a 16-core CPU, a single RTX3090 GPU, and 64GB RAM. The time overhead for one training step was calculated and averaged over 500 iterations. Due to the lack of optimization for parallel acceleration in the variance calculation code, which employs a simple 'for' loop approach, the actual overhead is larger than theoretically expected. Nonetheless, DISAM's advantage lies in its overhead being unrelated to gradient size, but only to batch size and domain number. This drawback can be addressed through parallel code optimization, and no additional memory overhead is present.

## C.5 DETAILS ABOUT CONVERGENCE CURVES OF SAM AND ERM

In this section, we provide a detailed analysis of the convergence curves depicted in Figure 1(b). Figure 7(a) presents the same as Figure 1(b), with a normalized representation of the loss curves,

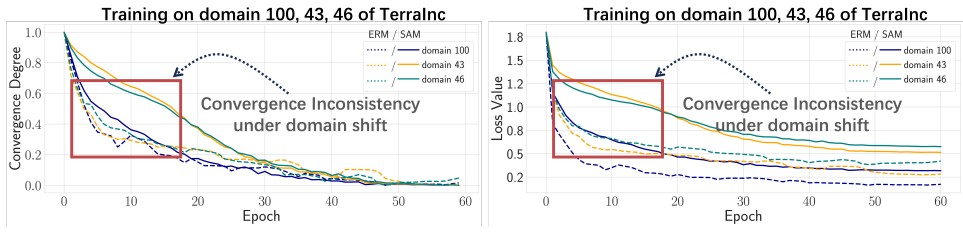

(a) Convergence curves under domain shifts    (b) Loss curves under domain shifts

Figure 7: Illustration of SAM's degradation of the training process under domain shifts. (a) Convergence curves of SAM and ERM for each domain during training, with the convergence degree normalized to [0,1]. (b) Loss curves of SAM and ERM for each domain during training.

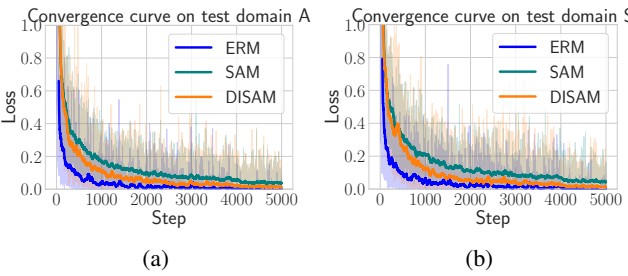

(a)                                    (b)

Figure 8: **Convergence curves** for ERM, SAM and DISAM. (a) & (b) show the trend of $\mathcal{L}(w)$ during the training process on PACS dataset.

ranging from 0 to 1, achieved by subtracting the minimum loss value and dividing by the maximum loss value. Our intention is to emphasize the inconsistency in convergence trends across different SAM domains, as illustrated by the optimization overshoot observed in Figure 7(a). Figure 7(b) showcases the actual loss change curve. It is apparent that due to the consistency issue encountered during the early phase of convergence, the in-domain convergence is compromised, resulting in poor generalization performance in the out-of-domain scenario.

## C.6 DETAILED ANALYSIS ABOUT OPEN-CLASS GENERALIZATION

In the experiments of open-class generalization, as presented in Table 3 and Figure 4 of section 4.4, we specifically explore the effectiveness of DISAM for parameter-efficient fine-tuning (PEFT). Our quantitative analysis compares the performance of ERM, SAM, and DISAM in fine-tuning scenarios. As shown in Table 3, although CoOp and CLIPOOD perform better on base classes than zero-shot, their performance on new classes is worse than zero-shot. This suggests that the fine-tuned parameters overfit to the existing training data distribution from both the domain and class perspectives. This overfitting is particularly detrimental to the generalization of large VLM models, which often have feature representations too rich for the downstream task, especially when only a small number of parameters are fine-tuned. Figure 4 visualizes the change in performance trends during the training process, and we observe a trend where ERM initially performs well on base classes but then exhibits a decline on new classes, suggesting a collapse of the feature space onto the training data classes. Although SAM offers some relief from overfitting, its performance on new classes does not match zero-shot levels. In contrast, DISAM, by minimizing sharpness more effectively, shows improved performance on new classes, especially in domain shift scenarios.

## C.7 DETAILED ANALYSIS OF CONVERGENCE SPEED COMPARISON

We presented a comparison of the convergence speed with the inclusion of ERM in Figure 8. It can be observed that although DISAM converges much faster compared to SAM, the overall convergence speed is still slower than ERM due to the introduction of $\rho$.

## D  PSEUDO CODE OF DISAM

We present pseudo-code for DISAM using Python syntax. PyTorch is utilized as the deep learning experimental framework. The code for the optimizer in the SAM-based method can be referenced from the provided open source links in the relevant papers.

Listing 1: Training Code for DISAM

```python
def train_epoch_disam(dataloader, model, optimizer):
    """
    Train the DISAM model for one epoch.

    Args:
        dataloader (DataLoader): The training dataloader.
        model (nn.Module): The training model.
        optimizer (Optimizer): The SAM-based optimizer, such as SAM,
            GSAM, and SAGM.
    """
    model.train()
    for i, data_list in tqdm(enumerate(dataloader)):
        imgs, labels = data_list
        imgs, labels = imgs.cuda(), labels.cuda()
        preds = model(imgs)
        # Calculate domain losses and total loss using the
            cross-entropy loss function
        domain_loss_list, total_loss = get_domain_loss(preds, labels,
            domain_labels, loss_func)
        loss_variance = compute_variance(domain_loss_list)
        loss = total_loss - lamda * loss_variance
        optimizer.zero_grad()
        loss.backward()
        # Perform the first step of SAM: gradient ascent with a fixed
            length rho
        optimizer.first_step(zero_grad=True)

        output = model(imgs)
        loss = loss_func(output, labels)
        loss.backward()
        # Obtain the actual gradient from the perturbation location of
            DISAM
        optimizer.second_step(zero_grad=True)

def get_domain_loss(preds, labels, domain_labels, loss_func):
    """
    The function to compute the loss for each domain.

    Args:
        preds (Tensor): The predictions of the training model in one
            batch.
        labels (Tensor): The labels of batch data.
        domain_labels (Tensor): The domain labels of batch data.
        loss_func: (Function): The loss function.
    """
    # Get a list of all domains
    domain_list = list(set(domain_labels))
    domain_loss_list = []
    total_loss = 0.

    for domain_name in domain_list:
        # Get the mask for the current domain
        domain_mask = domain_labels == domain_name

        labels_per_domain = labels[domain_mask]
```

```python
        preds_pre_domain = preds[domain_mask]

        # Compute the loss for the current domain
        single_domain_loss = loss_func(preds_pre_domain,
            labels_per_domain)

        domain_loss_list.append(single_domain_loss)

        # Add the loss for the current domain to the total loss, taking
            into account the number of samples in the domain
        total_loss += len(labels_per_domain) * single_domain_loss

    total_loss /= len(labels)

    return domain_loss_list, total_loss

def compute_variance(domain_loss_list):
    """
    The function to compute the variance of the list of domain losses

    Args:
        domain_loss_list (List): the list of each domain's loss.
    """
    loss_variance = 0.
    for domain_i_loss in domain_loss_list:
        for domain_j_loss in domain_loss_list:
            # Compute the square of the difference in loss between each
                pair of elements and add it to the loss variance
            loss_variance += (domain_i_loss - domain_j_loss)**2

    loss_variance /= (2*len(domain_loss_list)**2)

    return loss_variance
```

