# OpenReview forum: "Domain-Inspired Sharpness-Aware Minimization Under Domain Shifts"
_ICLR.cc/2024/Conference — ICLR 2024 poster_

### Official Review · Reviewer_BeTH · 2023-10-22

**Soundness:** 3 good
**Presentation:** 3 good
**Contribution:** 2 fair
**Rating:** 6
**Confidence:** 4

**Summary:**

Targeting at domain generalization scenario with possible shifts among domains, this paper proposes to take 'per domain optimality' into consideration for finding the perturbation of SAM. The proposed DISAM is shown to have an improved convergence rate. Numerically, DISAM outperforms other SAM alternatives.

**Strengths:**

S1. The idea of tackling domain shift in SAM is novel.

S2. A new algorithm, DISAM, is proposed with satisfying numerical results. DISAM improves over state-of-the-art by a large margin.

**Weaknesses:**

W1. Stronger motivation needed. The authors motivates the domain difference using Fig. 1 (b). While the convergence behaviors among domains are indeed inconsistent at the early stage,  the losses are similar after e.g., 30 epoch. The authors should also explain why the difference of convergence in **early phase** impact the generalization of SAM.


W2. More discussions on $\lambda$ in eq. (7) are needed. This is a critical parameter that considers the variance/domain shifts in DISAM. However, this $\lambda$ does not appear in Theorem 1. Can the authors illustrate more on this point? And how does the choice of $\lambda$ influence convergence and generalization?

**Questions:**

Q1. Relation with a recent work (https://arxiv.org/abs/2309.15639).

The paper above also proposes approaches to reduce variance for finding perturbations, although not designed for the domain generalization setting. How does this work relate with the proposed DISAM?


Q2. Theorem 1 illustrates that the *convergence* of DISAM benefits from $\Gamma$. Can the authors explain more on the discussion of
> as DISAM enjoys a smaller $\Gamma$ than SAM, DISAM can permit the potential larger $\rho$ than that in SAM, thus yielding a better generalization

In particular, how does the convergence rate link with generalization?

Q3. The last sentence in Sec 3 claims that
>  ... allowing larger $\rho$ for better generalization.

Why does larger $\rho$ relate to better generalization?

Q4. (minor) The notation in e.g., eq (5) can be improved, because the multiple subscripts $i$ in $\Sigma_{i} \frac{C_i}{\sum_i C_i}$ are confusing.

---

> ### Author Response · Authors · 2023-11-17
> **Response to Reviewer BeTH [1/2]**
>
> ## W1
>
> > Stronger motivation needed. The authors motivates the domain difference using Fig. 1 (b). While the convergence behaviors among domains are indeed inconsistent at the **early stage**, the losses are similar after e.g., 30 epoch. The authors should also explain why the difference of convergence in early phase impact the generalization of SAM.
>
> We would like to kindly point out that the y-axis of Figure.1 (b) represents the convergence degree instead of the loss value, which is computed by using the current loss value to divide the converged maximum loss. For the details, **we follow the reviewer's advice and have added one section to strenghen the explanation about the motivation in the Appendix C.5 of the revised version**.
>
> **Relationship of early-stage convergence and generalization:** In Figure 7(b) of page 24, we present the curves of various domain losses during training, which can promote the understanding. Basically, the inconsistency of the convergence degree in the early phase in Figure 7(a) **actually impairs the overall convergence of the model**. As can be seen in Figure 7(b), SAM has converged at a higher loss value, which reflects that the model is optimized towards a poorer local minima, resulting in the worse generalization performance.
>
>
> ## W2
>
> > More discussions on $\lambda$ in Eq. (7) are needed. This is a critical parameter that considers the variance/domain shifts in DISAM. However, this $\lambda$ does not appear in Theorem 1. Can the authors illustrate more on this point? And how does the choice of $\lambda$ influence convergence and generalization?
>
> Thank you very much for the reviewer's advice. We have elaborated more discussion about the role of $\lambda$ in DISAM in Appendix B.2 (pages 18-21) in the revised submission. In the proof of Theorem 1, specifically Eq. (15) on page 20, **$\lambda$ is integrated into $\beta$, serving as a hyperparameter that regulates the weight adjustment in DISAM**. It functions by modulating the degree of correction for domain shifts:
> $$
> \beta^i_t = \alpha_i - \frac{2\lambda}{M} \left (\mathcal{L}^i(w_t) - \frac{1}{M} \sum_{j=1}^M \mathcal{L}^j(w_t) \right)
> $$
>
> **The influence of $\lambda$:** The choice of $\lambda$ influences how aggressively DISAM responds to variance or domain shifts, with **a higher $\lambda$ leading to more pronounced adjustments in $\beta$**. Our experimental analysis in Figures 5(c) and (d) on page 9, reveals that DISAM's performance remains relatively stable across a wide range of $\lambda$ values. However, **choosing too large $\lambda$ can result in overly aggressive early training adjustments, yielding the negative impact on the convergence process and leading to the increased variance in repeated experiments**. Consequently, we adopted a default $\lambda$ value of 0.1 in all experiments.
>
> ## Q1
>
> > Relation with a recent work (https://arxiv.org/abs/2309.15639). The paper above also proposes approaches to reduce variance for finding perturbations, although not designed for the domain generalization setting. How does this work relate with the proposed DISAM?
>
> Thank you for recommending this excellent work on variance suppression (VaSSO)[1]. We have added this method into the revised submission with the proper discussion. Generally, while DISAM and VaSSO both enhance the perturbation direction generation in SAM, they target different aspects:
>
> - **VaSSO Approach:** VaSSO aims to **reduce noise from mini-batch sampling** by using averaged previous perturbation directions. Its primary focus is on **stabilizing perturbation generation within the same domain**.
>
> - **DISAM's Unique Focus:** In contrast, DISAM incorporates domain information to **specifically address domain-level convergence inconsistencies**, a challenge prevalent in domain shift scenarios. DISAM's approach is to **impose a variance minimization constraint on domain loss** during the perturbation generation process, thereby enabling a more representative perturbation location and enhancing generalization.
>
> We are running the experiments to compare/combine DISAM with VaSSO. Once the experiments are finished, we will report the results here and include the results in the submission.
>
> **Reference:**
>
> [1]. Enhancing Sharpness-Aware Optimization Through Variance Suppression, arXiv2023.

---

> > ### Author Response · Authors · 2023-11-21
> > **Response to Q1: Experiments to compare/combine DISAM with VaSSO.**
> >
> > ## Q1
> >
> > > Relation with a recent work (https://arxiv.org/abs/2309.15639). The paper above also proposes approaches to reduce variance for finding perturbations, although not designed for the domain generalization setting. How does this work relate with the proposed DISAM?
> >
> > We conduct experiments on the DomainBed benchmark to compare VaSSO[1] and DISAM.
> > - VaSSO **achieves a significant improvement in in-domain convergence by reducing perturbation direction noise** in PACS, VLCS, and TerraInc datasets. However, since it does not consider domain shift explicitly and only aims to make perturbation directions more consistent, it cannot guarantee that the perturbation directions are more representative, leading to a relatively modest improvement in out-of-domain performance.
> > - **DISAM demonstrates better generalization by incorporating domain information during perturbation direction generation**. Nonetheless, DISAM exhibits reduced convergence performance within the target domain on certain datasets, such as VLCS.
> > - **By combining DISAM and VaSSO, we can achieve synchronized improvements in both in-domain and out-of-domain performance.**
> >
> >
> > [**Table 1:** In-domain results]
> > | **Method**      | **PACS** | **VLCS** | **OfficeHome** | **TerraInc** | **DomainNet** | **Avg.** |
> > | --------------- | -------- | -------- | -------------- | ------------ | ------------- | -------- |
> > | SAM             | 97.3     | 84.8     | 85.8           | 88.9         | 68.5          | 85.1     |
> > | VaSSO[1]        | 97.8     | 85.7     | 86.4           | 94.5         | 68.6          | 86.6     |
> > | DISAM           | 97.8     | 84.4     | 86.3           | 94.8         | 70.2          | 86.7     |
> > | **VaSSO+DISAM** | 97.9     | 85.8     | 86.6           | 94.9         | 70.1          | **87.1** |
> >
> >
> > [**Table 2:** Out-of-domain results]
> > | **Method**      | **PACS** | **VLCS** | **OfficeHome** | **TerraInc** | **DomainNet** | **Avg.** |
> > | --------------- | -------- | -------- | -------------- | ------------ | ------------- | -------- |
> > | SAM             | 85.8     | 79.4     | 69.6           | 43.3         | 44.3          | 64.5     |
> > | VaSSO[1]        | 86.1     | 79.9     | 70.5           | 46.1         | 44.8          | 65.5     |
> > | DISAM           | 87.3     | 80.1     | 70.7           | 47.9         | 45.8          | 66.4     |
> > | **VaSSO+DISAM** | 87.2     | 80.2     | 70.9           | 48.0         | 45.8          | **66.5** |

---

> ### Author Response · Authors · 2023-11-17
> **Response to Reviewer BeTH [2/2]**
>
> ## Q2 & Q3
>
> > Q2: Theorem 1 illustrates that the convergence of DISAM benefits from $\Gamma$. Can the authors explain more on the discussion of as DISAM enjoys a smaller $\Gamma$ than SAM, DISAM can permit the potential larger $\rho$ than that in SAM, thus yielding a better generalization". In particular, how does the convergence rate link with generalization?
> > Q3. The last sentence in Sec 3 claims that ".. allowing larger $\rho$ for better generalization." Why does larger $\rho$ relate to better generalization?
>
> Thank you for the advice on explanation about convergence and generalization, and we have added more discussion and analysis in the Appendix B.3 on page 21. In the following, we provide some clarification on these points.
>
> - **Generalization Theorem of SAM:** In the SAM framework, the parameter $\rho$ plays a crucial role in determining generalizability. As established in [1], there exists an upper bound on the generalization error for SAM, suggesting that a larger $\rho$ could potentially enhance generalization, provided that convergence is not impeded. Here is the relevant generalization bound from [1]:
> > For any $\rho > 0$ and any distribution $\mathcal{D}$, with probability $1- \delta$ over the choice of the training set $S\sim \mathcal{D}$,
> > $$\mathcal{L} _{\mathcal{D}} (w) \leq \max _{\| \epsilon\| _2 \leq \rho} \mathcal{L} _{S}(w+\epsilon) + \sqrt{\frac{k \log \left(  1 + \frac{\| w \| _2^2}{\rho^2} (1+\sqrt{\frac{\log (n)}{k}})^2 \right) + 4 \log \frac{n}{\delta} + \tilde{O}(1)}{n-1}} $$
> > where $n = |S|$, $k$ is the number of parameters and we assumed $\mathcal{L} _{\mathcal{D}}(w) \leq \mathbb{E} _{\epsilon_i \approx \mathcal{N}(0, \rho)} [\mathcal{L} _{\mathcal{D}}(w+\epsilon)]$. DISAM leverages a smaller $\Gamma$ than SAM, as shown in Theorem 1 in page 5. This allows DISAM to employ a potentially larger $\rho$, enhancing generalizability.
>
> - **Practical Implications:** Combining the above theorem with the convergence theorem (Theorem 1 on page 5), there is a trade-off with respect to $\rho$. A larger $\rho$ might theoretically enhance generalization but poses greater challenges for convergence. This reflects the intuitive notion that searching for flatter minima across a broader range is inherently more complex, which can potentially affect training efficiency. However, **if $\mathcal{L}_{S} (w + \epsilon)$ can be converged with a sufficiently small value, a larger $\rho$ corresponds to better generalization**.  DISAM, with a smaller $\Gamma$ compared to SAM, converges faster, which means that under the same convergence speed, a larger $\rho$ can be used to achieve better generalization.
>
>
> - **Empirical Validation:** Our experiments, as illustrated in Figures 3(c) and (d) on page 6, demonstrate that DISAM effectively employs a larger $\rho$ compared to traditional SAM. DISAM's ability to handle a larger $\rho$ results in both consistent convergence and improved generalization compared to SAM, demonstrating its superiority in domain shift scenarios.
>
> We appreciate the reviewer's comments on the theoretical parts and have adjusted the corresponding parts to make these points more clear.
>
> **Reference:**
>
> [1]. Sharpness-aware minimization for efficiently improving generalization, ICLR2021.
>
>
> ## Q4
>
> > (minor) The notation in e.g., eq (5) can be improved, because the multiple subscripts $i$ in $\sum_{i}\frac{C_i}{\sum_{i}C_i}$ are confusing.
>
> We appreciate your detailed feedback on the notation in Eq. (5) and have updated it on page 4 to avoid confusion.

---

> ### Author Response · Authors · 2023-11-21
> **Kind Reminder to Respond Our Rebuttal**
>
> Dear Reviewer BeTH,
>
> We sincerely appreciate the effort and time you have devoted to providing constructive reviews, as well as your positive evaluation of our submission. As the deadline for discussion and paper revision is approaching,  we would like to offer a brief summary of our responses and updates:
>
> - Supplementary explanation for Figure 1(b)
> - The role and principles of $\lambda$ in the DISAM.
> - Similarities and differences between DISAM and VaSSO.
> - Mechanism of $\rho$'s impact on convergence and generalization
>
> **Would you mind checking our responses and confirming if you have any additional questions? We welcome any further comments and discussions!**
>
> Best Regards,
>
> The authors of Submission 1547

---

> > ### Author Response · Authors · 2023-11-23
> > **We anticipate your feedback!**
> >
> > Dear Reviewer BeTH,
> >
> > Thanks very much for your time and valuable comments.
> >
> > We understand you might be quite busy. However, the discussion deadline is approaching, and we have only a few hours left.
> >
> > Would you mind checking our response and confirming whether you have any further questions?
> >
> > Thanks for your attention.
> >
> > Best regards,
> >
> > The authors of submission 1547.

---

### Official Review · Reviewer_d1EH · 2023-10-26

**Soundness:** 4 excellent
**Presentation:** 4 excellent
**Contribution:** 3 good
**Rating:** 6
**Confidence:** 5

**Summary:**

The paper introduces the Domain-Inspired Sharpness Aware Minimization (DISAM) algorithm, a novel approach for optimizing under domain shifts. The motivation behind DISAM is to address the issue of inconsistent convergence rates across different domains when using Sharpness Aware Minimization (SAM), which can lead to optimization biases and hinder overall convergence.

The key innovation of DISAM lies in its focus on maintaining consistency in domain-level convergence. It achieves this by integrating a constraint that minimizes the variance in domain loss. This strategy allows for adaptive gradient perturbation: if a domain is already well-optimized (i.e., its loss is below the average), DISAM will automatically reduce the gradient perturbation for that domain, and increase it for less optimized domains. This approach helps balance the optimization process across various domains.

Theoretical analysis provided in the paper suggests that DISAM can lead to faster overall convergence and improved generalization, especially in scenarios with inconsistent domain convergence. The paper supports these claims with extensive experimental results, demonstrating that DISAM outperforms several state-of-the-art methods in various domain generalization benchmarks. Additionally, the paper highlights the efficiency of DISAM in fine-tuning parameters, particularly when combined with pretraining models, presenting a significant advancement in the field.

**Strengths:**

As of now, there has not yet been a sharpness-aware minimization (SAM) methodology developed specifically for addressing distribution shifts. The issue of varying convergence rates across different domains, as observed in SAM, is undeniably a significant challenge.

This methodology presents an impressive degree of compatibility, as it can be integrated with a variety of sharpness-variants. An especially commendable aspect of this approach is its computational efficiency. Compared to standard SAM techniques, it does not incur additional computational costs, making it a practical option for scenarios where resource constraints are a consideration.

In summary, the development of a SAM methodology that is adept at handling distribution shifts, and particularly its implications for domain convergence, is both novel and highly relevant in the current landscape of optimization challenges.

**Weaknesses:**

The idea of minimizing the variance between losses, a core aspect of the presented methodology, is not entirely novel. Similar concepts have been previously explored in methods like vREX (Out-of-Distribution Generalization via Risk Extrapolation) and further extended to gradient computations in methodologies like Fishr (Invariant Gradient Variances for Out-of-Distribution Generalization). In this context, the proposed approach appears to be an incremental adaptation of vREX principles applied specifically to the challenges faced in Sharpness Aware Minimization (SAM) scenarios.

The improvement in out-of-distribution (OOD) performance using the DISAM methodology does not appear intuitive. In fact, when comparing its performance enhancements to those achieved with CLIPOOD, as reported, the difference seems marginal. This observation raises questions about the actual effectiveness of DISAM, particularly in the context of fine-tuning methodologies.

**Questions:**

Similar to how transitioning from ERM to vREX in optimization has been shown to enhance domain generalization performance, the application of vREX to SAM in the form of this methodology could be seen as a natural extension that brings comparable performance improvements. Furthermore, it is a valid assertion that incorporating various algorithms tailored for domain generalization (such as Fish, Fishr, gradient alignment) into the SAM optimization framework could potentially yield performance enhancements. The logic here is that these methods, when applied within the context of SAM, could enhance its ability to generalize across domains.

However, the critique that DISAM may simply be an incremental version of applying domain generalization methodologies to SAM is not without its counterarguments. It's important to consider the specific challenges and nuances of the SAM framework and how DISAM addresses these. If DISAM introduces significant modifications or adaptations that are uniquely tailored to the idiosyncrasies of SAM, then its contribution could extend beyond a mere incremental update. The key would lie in the specifics of how DISAM modifies or enhances the existing principles of SAM and domain generalization methods, making it more than just a straightforward application of known techniques.

In summary, while the perspective that DISAM is an incremental version of existing methodologies is certainly tenable, a comprehensive evaluation would require a deeper exploration of how DISAM specifically adapts or augments the SAM framework to address its unique challenges. If such adaptations are significant, they could justify the novelty and utility of DISAM beyond a simple combination of existing techniques.

Can you provide the reproducible code during the rebuttal period?

**Details Of Ethics Concerns:**

\

---

> ### Author Response · Authors · 2023-11-17
> **Response to Reviewer d1EH [1/3]**
>
> ## W1 & Question
>
> > W1: The idea of minimizing the variance between losses, a core aspect of the presented methodology, is not entirely novel. Similar concepts have been previously explored in methods like vREX (Out-of-Distribution Generalization via Risk Extrapolation) and further extended to gradient computations in methodologies like Fishr (Invariant Gradient Variances for Out-of-Distribution Generalization). In this context, the proposed approach appears to be an incremental adaptation of vREX principles applied specifically to the challenges faced in Sharpness Aware Minimization (SAM) scenarios.
> > Question: In summary, while the perspective that DISAM is an incremental version of existing methodologies is certainly tenable, a comprehensive evaluation would require a deeper exploration of how DISAM specifically adapts or augments the SAM framework to address its unique challenges. If such adaptations are significant, they could justify the novelty and utility of DISAM beyond a simple combination of existing techniques.
>
> First, we are sorry about one typo issue in Eq.(7) that may mislead the understanding of the reviewer, which we has corrected with the proper description in the revised submission. **Concretely, the $w$ (i.e., $\hat{w}$ in the revised version) in the variance term is actually without derivative taken, when optimzing the model parameter**. That is to say, the $w$ (i.e., $\hat{w}$ in the revised version) only makes effect during the inner loop for the perturbation generation. This makes DISAM intrinsically different from V-REx [1] (an extension of IRM [2]). We use a table in the following the comprehensively compare DISAM and VRex to clarify our difference. Generally, V-REx focuses on **achieving consistent loss values across domains**, and Fish [3] and Fishr [4] **emphasize gradient consistency across domains** to foster out-of-domain generalization.
>
> In comaprison, DISAM targets to address the challenge of **effective perturbation directions for sharpness estimation in domain shift scenarios** by introducing the guidance of the domain-level loss variance minimization, which actually does not affect the training objective (i.e., the first term of Eq.(7)). Unlike V-REx, which directly minimizes domain loss variance and can negatively impact convergence, or Fish and Fishr, which constrain gradient updates, **DISAM adopts a distinct strategy, minimizing sharpness of multiple domains consistently, to enhance generalization**.
>
> |  Method   |    Total Optimization Function | Optimization on $w$   | Optimization on $\epsilon$       |
> | -------- |-------------|-------------|--------------------|
> | ERM      | $\min _{w} \sum _{i=1}^M  \alpha _i \mathcal{L} _{i}(w)$ | Same to left | $\times$ |
> | V-REx[1] | $\min _{w} \sum _{i=1}^M \alpha _i \mathcal{L} _{i}(w) + \beta \text{Var} \{ \mathcal{L} _i(w) \} _{i=1}^M$    | Same to left                  | $\times$    |
> | Fish[3]  | $\min _{w} \sum _{i=1}^M  \alpha _i \mathcal{L} _{i}(w) - \gamma \frac{2}{M(M-1)} \sum _{i,j \in [1,M]}^{i \neq j} \nabla \mathcal{L} _{i}(w) \cdot \nabla \mathcal{L} _{j}(w)$       | Same to left   | $\times$ |
> | Fishr[4] | $\min _{w} \sum _{i=1}^M  \alpha _i \mathcal{L} _{i}(w) - \lambda \frac{1}{M} \sum _{i=1}^{M} \| \nabla \mathcal{L} _{i}(w) - \nabla \mathcal{L}(w)\|^2$                             | Same to left | $\times$ |
> | SAM      | $\min _{w}  \max _{\| \epsilon\|_2 \leq \rho}  \sum _{i=1}^M \alpha _i \mathcal{L} _i(w + \epsilon)$ | $\min _{w}  \sum _{i=1}^M \alpha _i \mathcal{L} _i(w + \epsilon)$    | $\max _{\| \epsilon\|_2 \leq \rho}  \sum _{i=1}^M \alpha _i \mathcal{L} _i(w + \epsilon)$      |
> | DISAM    | $\min _{w}  \max _{\| \epsilon\|_2 \leq \rho} \left[ \sum _{i=1}^M \alpha _i \mathcal{L} _i(w + \epsilon) -  \lambda \text{Var}\{\mathcal{L} _i(\hat{w} + \epsilon)\} _{i=1}^M \right]$ | $\min _{w}  \sum _{i=1}^M \alpha _i \mathcal{L} _i(w + \epsilon)$  | $\max _{\| \epsilon\|_2 \leq \rho} \left[ \sum _{i=1}^M \alpha _i \mathcal{L} _i(w + \epsilon) -  \lambda \text{Var}\{\mathcal{L} _i(w + \epsilon)\} _{i=1}^M \right]$ |

---

> ### Author Response · Authors · 2023-11-17
> **Response to Reviewer d1EH [2/3]**
>
> ## W1 & Question
> > W1: The idea of minimizing the variance between losses, a core aspect of the presented methodology, is not entirely novel. Similar concepts have been previously explored in methods like vREX (Out-of-Distribution Generalization via Risk Extrapolation) and further extended to gradient computations in methodologies like Fishr (Invariant Gradient Variances for Out-of-Distribution Generalization). In this context, the proposed approach appears to be an incremental adaptation of vREX principles applied specifically to the challenges faced in Sharpness Aware Minimization (SAM) scenarios.
> > Question: In summary, while the perspective that DISAM is an incremental version of existing methodologies is certainly tenable, a comprehensive evaluation would require a deeper exploration of how DISAM specifically adapts or augments the SAM framework to address its unique challenges. If such adaptations are significant, they could justify the novelty and utility of DISAM beyond a simple combination of existing techniques.
>
>
>
> Second, DISAM is orthogonal to existing state-of-the-art methods including V-REx and Fishr and can improve their performance regarding generalization. The following tables present some results to prove DISAM’s superiority:
>
> [**Table 1.** Comparison with existing methods.]
> | **Method**      | **PACS** | **VLCS** | **OfficeHome** | **TerraInc** | **DomainNet** | **Avg.** |
> | --------------- | :--------: | :--------: | :--------------: | :------------: | :-------------: | :--------: |
> | IRM             | 83.5     | 78.5     | 64.3           | 47.6         | 33.9          | 61.6     |
> | V-REx[1]         | 84.9     | 78.3     | 66.4           | 46.4         | 33.6          | 61.9     |
> | **V-REx+DISAM**  | 85.8     | 78.4     | 70.5           | 45.9         | 42.3          | **64.6**     |
> | Fish[3]         | 85.5     | 77.8     | 68.6           | 45.1         | 42.7          | 63.9     |
> | Fishr[4]        | 86.9     | 78.2     | 68.2           | 53.6         | 41.8          | 65.7     |
> | **Fishr+DISAM** | 87.5     | 79.2     | 70.7           | 54.8         | 43.9          | **67.2** |
>
>
>
> [**Table 2.** Comparison with existing SAM-based methods.]
> | **Method**     | **PACS** | **VLCS** | **OfficeHome** | **TerraInc** | **DomainNet** | **Avg.** |
> | --------------- | :--------: | :--------: | :--------------: | :------------: | :-------------: | :--------: |
> | SAM            | 85.8     | 79.4     | 69.6           | 43.3         | 44.3          | 64.5     |
> | **SAM+DISAM**  | 87.3     | 80.1     | 70.7           | 47.9         | 45.8          | **66.4** |
> | GSAM           | 85.9     | 79.1     | 69.3           | 47.0         | 44.6          | 65.1     |
> | **GSAM+DISAM** | 87.2     | 80.0     | 70.8           | 50.6         | 45.6          | **66.8** |
> | SAGM           | 86.6     | 80.0     | 70.1           | 48.8         | 45.0          | 66.1     |
> | **SAGM+DISAM** | 87.5     | 80.7     | 71.0           | 50.0         | 46.0          | **67.0** |
>
> These results clearly demonstrate DISAM’s distinctive approach and its effectiveness in enhancing generalization, supporting its novelty and utility beyond existing methodologies.
>
> We appreciate the reviewer's constructive suggestion, and added a comparison table and more discussion for related works (see Appendix A.2.5 on page 17-18) to clarify the novelty of DISAM in the revision.
>
> **Reference:**
>
> [1]. Out-of-distribution generalization via risk extrapolation (rex), ICML2021.
>
> [2]. Invariant risk minimization, arXiv2019.
>
> [3]. Gradient matching for domain generalization, arXiv2021.
>
> [4]. Fishr: Invariant gradient variances for out-of-distribution generalization, ICML2022.

---

> ### Author Response · Authors · 2023-11-17
> **Response to Reviewer d1EH [3/3]**
>
> ## W2
>
> > The improvement in out-of-distribution (OOD) performance using the DISAM methodology does not appear intuitive. In fact, when comparing its performance enhancements to those achieved with CLIPOOD, as reported, the difference seems marginal. This observation raises questions about the actual effectiveness of DISAM, particularly in the context of fine-tuning methodologies.
>
> We would like to kindly clarify that in Table 2, the results of CLIPOOD in gray are from the orginal paper [1], but we cannot reproduce these results with the open source code of [1] (even after the considerable hyparameter searching). **The best results with their open source code are reported as CLIPOOD*** and on the basis of these results, DISAM's improvement is actually not marginal (see the following table for convenience).
>
> Besides, it's crucial to highlight that **DISAM's advantages become more evident in open-class scenarios**. As can be seen in the following table (or Table 3 on page 8), DISAM notably outperforms CLIPOOD in these scenarios, achieving an average improvement of 2.8% on new classes and 1.0% on base classes. This is significant, considering that CoOp and CLIPOOD even underperform compared to zero-shot results in new classes.
>
> | **Method** | **Results on DomainBed** | **Open-class Results on Base Classes** | **Open-class Results on New Classes** |
> | ---------- | :------------------------: | :------------------------------: | :-----------------------------: |
> | Zero-shot  | 70.2                     | 72.6                           | 67.4                          |
> | CoOp       | 73.4                     | 74.4                           | 66.3                          |
> | **+DISAM** |**74.8**                    | **75.3**                       | **69.6**                      |
> | CLIPOOD*    | 77.9                      | 76.0                           | 66.9                          |
> | **+DISAM** | **78.8**                       | **77.0**                       | **69.7**                      |
>
> For a comprehensive understanding of how DISAM enhances fine-tuning in open-class scenarios, we kindly refer the reviewer to the in-depth analysis provided in Appendix C.6 on page 25. This section substantiates DISAM's role in improving generalization in fine-tuning, especially in contexts that existing methods are challenging to achieve improvement.
>
> **Reference:**
>
> [1]. CLIPOOD: Generalizing CLIP to Out-of-Distributions, ICML2023.
>
>
> ## Reproducible code
>
> > Can you provide the reproducible code during the rebuttal period?
>
> We provide an anonymized version of the code repository, accessible through this 2-hop link: [https://openreview.net/forum?id=I4wB3HA3dJ&noteId=e1Uu30vHqy]. To promote the understanding of the code, the reviewers can also combine with Algorithm 1 and the "Pseudo Code of DISAM" in Appendix D on page 25-26.

---

> ### Author Response · Authors · 2023-11-21
> **Kind Reminder to Respond Our Rebuttal**
>
> Dear Reviewer d1EH,
>
> We sincerely appreciate the effort and time you have devoted to providing constructive reviews, as well as your positive evaluation of our submission. As the deadline for discussion and paper revision is approaching,  we would like to offer a brief summary of our responses and updates:
>
> - Detailed explanation of DISAM's novelty and contributions.
> - Description and analysis in an open-class setting for DISAM.
> - Submission of reproducible code.
>
> **Would you mind checking our responses and confirming if you have any additional questions? We welcome any further comments and discussions!**
>
> Best Regards,
>
> The authors of Submission 1547

---

> > ### Author Response · Authors · 2023-11-23
> > **Official Comment by Authors**
> >
> > We greatly appreciate the reviewer's dedicated time and effort in thoroughly reviewing our paper and providing professional and valuable feedback.
> >
> > Best,
> >
> > The authors of submission 1547.

---

### Official Review · Reviewer_mma9 · 2023-10-30

**Soundness:** 3 good
**Presentation:** 3 good
**Contribution:** 3 good
**Rating:** 8
**Confidence:** 4

**Summary:**

This paper introduces a novel optimization algorithm named Domain-Inspired Sharpness Aware Minimization (DISAM) tailored for challenges arising from domain shifts. It seeks to maintain consistency in sharpness estimation across domains by introducing a constraint to minimize the variance in domain loss. This approach facilitates adaptive gradient adjustments based on the optimization state of individual domains. Theoretical and empirical findings show the proposed method offers faster convergence and superior generalization under domain shifts.

**Strengths:**

1.	The proposed method targets at the model generalization under domain shifts, which is a common challenge in machine learning. To date, there has been a lack of thorough investigation into sharpness-based optimization in the context of domain shifts, and the idea of constraint the variance of losses among training domains is interesting.

2.	The paper not only presents theoretical evidence showcasing the efficiency of DISAM, but it also provides empirical data to support this claim, demonstrating the improved performance across various domain generalization benchmarks.

3.	The analytical experiments conducted in this paper are comprehensive and lucid, providing evidence of DISAM's efficacy in enhancing convergence speed and mitigating model sharpness. Additionally, the study investigates the application of DISAM for fine-tuning a clip-based model, aiming to achieve improved open-class generalization.

**Weaknesses:**

1.	SAM-based optimization incurs twice the computational overhead and additional storage overhead in comparison to the commonly used SGD. While DISAM, the method proposed in this paper, demonstrates faster convergence under domain shift conditions when compared to SAM, it does not include a comparison with optimizers such as SGD or Adam.

2.	This paper employs multiple benchmarks to evaluate the performance of multi-source domain generalization. The article highlights the need for advancements in the domain shift perspective of the SAM method and suggests conducting comparisons between DISAM and the state-of-the-art (SOTA) method to further validate the effectiveness of the proposed approach.

3.	The value of $\rho$ in DISAM significantly influences both the convergence speed and generalizability. And it needs more discussion on how to effectively determine the value to maximize the benefits of proposed method.

**Questions:**

1.	The article presents a theoretical analysis suggesting that larger values of parameter $\rho$ should lead to improved generalization, given that convergence is guaranteed. It is important to reflect this aspect in the experiments to provide stronger evidence and validation.

2.	Regarding the open class generalization of the clip-based model, further experimental analysis should be conducted to elucidate the reasons behind the superior performance of DISAM.

For other questions, please refer to the weaknesses.

---

> ### Author Response · Authors · 2023-11-17
> **Response to Reviewer mma9 [1/2]**
>
> ## W1
>
> > SAM-based optimization incurs twice the computational overhead and additional storage overhead in comparison to the commonly used SGD. While DISAM, the method proposed in this paper, demonstrates faster convergence under domain shift conditions when compared to SAM, it does not include a comparison with optimizers such as SGD or Adam.
>
> Thank you for highlighting the point of the computational cost. We have to admit that although SAM-based methods help  improve generalization, it is ususally at the expense of approximately double the computational effort per iteration compared to standard SGD, due to the sharpness-aware perturbation. DISAM shares the similar cost as other SAM-based methods, since we target to address the domain-level inconsistency during the sharpeness estimation, instead of the training accelaration. In the revision version, we **highlight this point with the empirical verification** in Figure 8 on page 25, pointing out that compared DISAM with ERM, revealing that although DISAM converges faster than SAM, it does not outpace ERM in terms of convergence speed. **The potential extension combined with the acceleration methods like [1,2] can be explored** in the future works.
>
> **Reference:**
>
> [1]. Efficient sharpness-aware minimization for improved training of neural networks, ICLR2022.
>
> [2]. Sharpness-aware training for free, NeurIPS2022.
>
>
>
> ## W2
>
> > This paper employs multiple benchmarks to evaluate the performance of multi-source domain generalization. The article highlights the need for advancements in the domain shift perspective of the SAM method and suggests conducting comparisons between DISAM and the state-of-the-art (SOTA) method to further validate the effectiveness of the proposed approach.
>
> We would like to kindly clarify that in Table 1, we has conducted comparison with some SOTA methods [1-2] like the latest SAM-based method SAGM [1] in combination with CORAL [2]. To further address the reviewer's concern, we here add two new SOTA methods "Decompose, Adjust, Compose"(DAC-SC) [3] and Fishr [4] (following reviewer d1EH's recommendation). The table below presents the performance of different methods across multiple domains:
> | **Method**            | **PACS** | **VLCS** | **OfficeHome** | **TerraInc** | **DomainNet** | **Avg.** |
> | --------------------- |:--------:|:--------:|:--------------:|:------------:|:-------------:|:--------:|
> | ERM                   |   85.5   |   77.5   |      66.5      |     46.1     |     43.8      |   63.9   |
> | SAM                   |   85.8   |   79.4   |      69.6      |     43.3     |     44.3      |   64.5   |
> | **DISAM**             |   87.3   |   80.1   |      70.7      |     47.9     |     45.8      | **66.4** |
> | Fishr (ICML2022) [4]  |   86.9   |   78.2   |      68.2      |     53.6     |     41.8      |   65.7   |
> | **Fishr+DISAM**       |   87.5   |   79.2   |      70.7      |     54.8     |     43.9      | **67.2** |
> | DAC-SC (CVPR2023) [3] |   87.5   |   78.7   |      70.3      |     46.5     |     44.9      |   65.6   |
> | **DAC-SC+DISAM**      |   88.7   |   79.1   |      70.6      |     47.4     |     45.6      | **66.3** |
> | SAGM (CVPR2023) [1]   |   86.6   |   80.0   |      70.1      |     48.8     |     45.0      |   66.1   |
> | **SAGM+DISAM**        |   87.5   |   80.7   |      71.0      |     50.0     |     46.0      | **67.0** |
>
> These results clearly illustrate the consistent improvement of DISAM on a range of methods in the field of multi-source domain generalization.
>
> **Reference:**
>
> [1]. Sharpness-aware gradient matching for domain generalization, CVPR2023.
>
> [2]. Deep coral: Correlation alignment for deep domain adaptation, ECCV2016.
>
> [3]. Decompose, Adjust, Compose: Effective Normalization by Playing with Frequency for Domain Generalization, CVPR2023.
>
> [4]. Fishr: Invariant gradient variances for out-of-distribution generalization, ICML2022.

---

> ### Author Response · Authors · 2023-11-17
> **Response to Reviewer mma9 [2/2]**
>
> ## W3 & Q1
>
> > W3: The value of $\rho$ in DISAM significantly influences both the convergence speed and generalizability. And it needs more discussion on how to effectively determine the value to maximize the benefits of proposed method.
> > Q1: The article presents a theoretical analysis suggesting that larger values of parameter $\rho$ should lead to improved generalization, given that convergence is guaranteed. It is important to reflect this aspect in the experiments to provide stronger evidence and validation.
>
> Thank you for the suggestion. Following the advice, we have enriched the discussion about the value of $\rho$ in the Appendix B.3 (page 21) of the revised version. Regarding the experiments, we actually have conducted the corresponding experiments to verify this aspect. We kindly refer the reviewer to Appendix B.3 for more details. We summarize the parts for the reviewer's questions as follows.
>
>
> - **Generalization Theorem of SAM:** In the SAM framework, the parameter $\rho$ plays a crucial role in determining generalizability. As established in [1], there exists an upper bound on the generalization error for SAM, suggesting that **a larger $\rho$ could potentially enhance generalization, provided that convergence is not impeded**. Here is the relevant generalization bound from [1]:
> > For any $\rho > 0$ and any distribution $\mathcal{D}$, with probability $1- \delta$ over the choice of the training set $S\sim \mathcal{D}$,
> > $$\mathcal{L} _{\mathcal{D}} (w) \leq \max _{\| \epsilon\| _2 \leq \rho} \mathcal{L} _{S}(w+\epsilon) + \sqrt{\frac{k \log \left(  1 + \frac{\| w \| _2^2}{\rho^2} (1+\sqrt{\frac{\log (n)}{k}})^2 \right) + 4 \log \frac{n}{\delta} + \tilde{O}(1)}{n-1}} $$
> > where $n = |S|$, $k$ is the number of parameters and we assumed $\mathcal{L} _{\mathcal{D}}(w) \leq \mathbb{E} _{\epsilon_i \approx \mathcal{N}(0, \rho)} [\mathcal{L} _{\mathcal{D}}(w+\epsilon)]$. This theorem's proof focuses solely on the magnitude of $\rho$, thus affirming the applicability of this theoretical framework to DISAM.
>
> - **Practical Implications:** When considering the convergence Theorem 1 on page 5 alongside the above generalization theorem, a critical trade-off emerges with respect to $\rho$. **A larger $\rho$ might theoretically enhance generalization but poses greater challenges for convergence**. This reflects the intuitive notion that searching for flatter minima across a broader range is inherently more complex, which can potentially affect training efficiency.
>
> - **Empirical Validation:** DISAM, with its accelerated convergence, can utilize a larger $\rho$ while still maintaining an acceptable convergence. This advantage is empirically showcased in Figures 3(c) and (d) on page 6, where we demonstrate that DISAM effectively employs a larger $\rho$ compared to traditional SAM. This ensures both convergence and enhanced generalization. Such a capability to balance between convergence efficiency and generalization is a distinguishing feature of DISAM over conventional SAM methods.
>
> **Reference:**
>
> [1]. Invariant risk minimization, arXiv2019.
>
>
> ## Q2
>
> > Regarding the open class generalization of the clip-based model, further experimental analysis should be conducted to elucidate the reasons behind the superior performance of DISAM.
>
> Thank you for the suggestion. To clarify the reasons, we complemented more analysis in Appendix C.6 on page 25 in the revised version. Generally, according to the results in Table 3 (we select some results in the following table for reference) and Figure 4, we can see:
>
> - **ERM tends to overfit to training data classes:** As shown in the table below, although CoOp and CLIPOOD perform better on base classes than zero-shot, their performance on new classes is worse than zero-shot. This suggests that the fine-tuned parameters **overfit to the existing training data distribution from both the domain and class perspectives**. Figure 4 visualizes the change in performance trends during the training process, and we observe a trend where ERM initially performs well on base classes but then exhibits a decline on new classes, suggesting a collapse of the feature space onto the training data classes.
>
> - **DISAM improves generalization on new classes:** Although SAM offers some relief from overfitting, its performance on new classes does not match zero-shot levels. In contrast, DISAM, by minimizing sharpness more effectively, shows improved performance on new classes, especially in domain shift scenarios.
>
>
> | **Method** | **Results on Base** | **Results on New** |
> | ---------- |:-------------------:|:------------------:|
> | Zero-shot  |        72.6         |        67.4        |
> | CoOp       |        74.4         |        66.3        |
> | **CoOp+DISAM** |      **75.3**       |      **69.6**      |
> | CLIPOOD    |        76.0         |        66.9        |
> | **CLIPOOD+DISAM** |      **77.0**       |      **69.7**      |

---

> ### Author Response · Authors · 2023-11-21
> **Kind Reminder to Respond Our Rebuttal**
>
> Dear Reviewer mma9,
>
> We sincerely appreciate the effort and time you have devoted to providing constructive reviews, as well as your positive evaluation of our submission. As the deadline for discussion and paper revision is approaching,  we would like to offer a brief summary of our responses and updates:
>
> - Comparison of convergence speed on ERM.
> - Expanded comparisons with more state-of-the-art methods.
> - Discussion on $\rho$'s impact on convergence and generalization.
> - Explanations for the open-class experiments.
>
> **Would you mind checking our responses and confirming if you have any additional questions? We welcome any further comments and discussions!**
>
> Best Regards,
>
> The authors of Submission 1547

---

> > ### Comment · Reviewer_mma9 · 2023-11-23
> > **Thanks for the reply.**
> >
> > Thanks for the response. After reading the authors’ thorough rebuttal and other reviewers’ comments, I feel the authors have well addressed my concerns and thus will increased my score.
> >
> > Best,
> > The reviewer

---

> > > ### Author Response · Authors · 2023-11-23
> > > **Thanks for your feedback and positive support!**
> > >
> > > We sincerely appreciate your feedback regarding our efforts to address your concerns, and we would like to express our gratitude for your positive support. We will carefully consider all of your advice and incorporate the resulting improvements into the final version.
> > >
> > > Best,
> > >
> > > The authors of submission 1547.

---

### Official Review · Reviewer_sVwJ · 2023-10-31

**Soundness:** 2 fair
**Presentation:** 3 good
**Contribution:** 2 fair
**Rating:** 5
**Confidence:** 3

**Summary:**

Due to the inconsistent convergence degree of SAM across different domains, the optimization may bias towards certain domains and thus impair the overall convergence. To address this issue, this paper considers the domain-level convergence consistency in the sharpness estimation to prevent the overwhelming perturbations for less optimized domains. Specifically, DISAM introduces the constraint of minimizing variance in the domain loss. When one domain is optimized above the averaging level w.r.t. loss, the gradient perturbation towards that domain will be weakened automatically, and vice versa.

**Strengths:**

They identify that the use of SAM has a detrimental impact on training under domain shifts, and further analyze that the reason is the inconsistent convergence of training domains that deviates from the underlying i.i.d assumption of SAM.

**Weaknesses:**

This paper considers the domain-level convergence consistency in SAM for multiple domains, and proposes to adopts the domain loss variance in training loss. The convergence consistency is a general issue, and the solution is normal, thus the novelty is not so clear for publication in ICLR.

**Questions:**

1.	In the definition of the variance between different domain losses, the values of loss between different domains are restricted. Which one is more import? The value of losses in different domains, or the minimization speed of loss in different domains?
2.	In the learning of multiple domains, there is Multi-Objective Optimization, so the domain-level convergence consistency is a general issue under domain shifts? Or the convergence consistency is a general issue in Multi-Objective Optimization?
3.	This paper considers the domain-level convergence consistency in SAM for multiple domains, and proposes to adopts the domain loss variance in training loss. The convergence consistency is a general issue, and the solution is normal, thus the novelty is not so clear.

---

> ### Author Response · Authors · 2023-11-17
> **Response to Reviewer sVwJ [1/2]**
>
> ## Weakness & Q3
>
> > Weakness: This paper considers the domain-level convergence consistency in SAM for multiple domains, and proposes to adopts the domain loss variance in training loss. The convergence consistency is a general issue, and the solution is normal, thus the novelty is not so clear for publication in ICLR.
> > Q3: This paper considers the domain-level convergence consistency in SAM for multiple domains, and proposes to adopts the domain loss variance in training loss. The convergence consistency is a general issue, and the solution is normal, thus the novelty is not so clear.
>
>
>
> **Differences from general convergence consistency issue:**
> - **Distinct focus:** DISAM focuses on the issue where **SAM-based methods are unable to accurately estimate sharpness in domain shift scenarios**, leading to the ineffective sharpness minimization and reduction in generalization performance.
> - **Enhancing on top of general methods:** While traditional solutions[1,2,3] aim at convergence consistency in parameter optimization, DISAM's methodology is distinct and orthogonal. It builds upon methods like V-REx[2] and Fishr[3], but **goes further in enhancing out-of-domain generalization through better sharpness minimization**. This is evident in our experiments, where combining DISAM with Fishr results in significant performance gains (as shown in the table below).
>
> | **Method**      | **PACS** | **VLCS** | **OfficeHome** | **TerraInc** | **DomainNet** | **Avg.** |
> | --------------- |:--------:|:--------:|:--------------:|:------------:|:-------------:|:--------:|
> | V-REx[2]        |   84.9   |   78.3   |      66.4      |     46.4     |     33.6      |   61.9   |
> | **V-REx+DISAM** |   85.8   |   78.4   |      70.5      |     45.9     |     42.3      |   64.6   |
> | Fishr[3]        |   86.9   |   78.2   |      68.2      |     53.6     |     41.8      |   65.7   |
> | **Fishr+DISAM** |   87.5   |   79.2   |      70.7      |     54.8     |     43.9      | **67.2** |
>
>
> **Novelty and contributions:**
> - We first identify that the straightforward application of SAM has **a detrimental impact on training under domain shifts** (as shown Figure 1 and Table 1). Specifically, we observed that the way SAM generates perturbation directions amplifies the inconsistency in convergence between domains, leading to inaccurate sharpness estimation and making sharpness minimization less effective.
> - DISAM handle the above challenge by imposing a variance minimization constraint on domain loss **during the sharpness estimation process**, thereby enabling a more representative perturbation location and enhancing generalization.
> - The **significant improvements in extensive experimental results** (as shown in Table 1-3) validate DISAM's novelty and practical relevance.
>
> **Reference:**
>
> [1]. Invariant risk minimization, arXiv2019.
>
> [2]. Out-of-distribution generalization via risk extrapolation (rex), ICML2021.
>
> [3]. Fishr: Invariant gradient variances for out-of-distribution generalization, ICML2022.

---

> ### Author Response · Authors · 2023-11-17
> **Response to Reviewer sVwJ [2/2]**
>
> ## Q1
>
> > In the definition of the variance between different domain losses, the values of loss between different domains are restricted. Which one is more import? The value of losses in different domains, or the minimization speed of loss in different domains?
>
>
> Firstly, we are sorry that one typo in Eq. (7) has mislead your understanding. In the revised manuscript, we have addressed such a typo issue in the description of Eq.(7) as follows.
>
> $$
> \min _{w} \mathbb{E}  _{\xi \in \mathcal{S}} [\mathcal{L} _{DISAM}(w;\xi)] \triangleq \min _{w}  \max _{ \|
>  \epsilon \| _2 \leq \rho} \left [ \sum _{i=1} ^M \alpha _i \mathcal{L} _i (w + \epsilon) -  \lambda \text{Var}\{\mathcal{L} _i(\hat{w} + \epsilon)\} _{i=1} ^M \right]
> $$
>
> Here **$\hat{w}$ is $w$ without derivative taken during optimization**, and it only makes effect in the $\max_{\| \epsilon\|_2 \leq \rho}$ loop without affecting the first term.
>
> - **Variance Minimization Focus:** DISAM primarily focuses on minimizing variance during the generation of perturbation directions $\epsilon$. The outer optimization w.r.t. $w$ does not involve a trade-off between the empirical loss term and the variance term as we enforce $\hat{w}$ (assigned by $w$) without derivative taken.
> - **Crucial Role of Minimizing Variance:** Minimizing domain-level variance in the perturbation generation loop is critical. Our experiments, illustrated in Figures 5(c) and 5(d) on page 9, show a marked decrease in generalization performance when $\lambda=0$, confirming its essential effectiveness in DISAM. Furthermore, DISAM exhibits a robust performance across a broad range of $\lambda$ values.
>
>
> ## Q2
>
> > In the learning of multiple domains, there is Multi-Objective Optimization, so the domain-level convergence consistency is a general issue under domain shifts? Or the convergence consistency is a general issue in Multi-Objective Optimization?
>
> **Differences from multi-objective optimization:**
> - The research topic is intrinsically different from multi-objective optimization. Specially, the goal of DISAM has a single ultimate objective, improving the generalization performance of the model trained under multiple domains. This emphasizes both in-domain generalization and out-of-domain generalization, while multi-objective optimization usually refers to improving the multiple objectives of all collaborated tasks.
> - Methodologically, we still use the SAM optimization framework, and **do not involve multi-objective optimization process during training**. Specifically, $\{\alpha_i=n_i/N\}_{i=1}^M$ are constant in Eq.(7) unlike the task-specific variables to be learnt in multi-objective optimization. We observed the negative impact of domain-level convergence inconsistency on SAM-based methods during the perturbation direction generation process. DISAM achieves better perturbation directions for out-of-domain generalization by minimizing the variance of the domain losses.

---

> ### Author Response · Authors · 2023-11-21
> **Kind Reminder to Respond Our Rebuttal**
>
> Dear Reviewer sVwJ,
>
> We sincerely appreciate the effort and time you have devoted to providing constructive reviews, as well as your positive evaluation of our submission. As the deadline for discussion and paper revision is approaching,  we would like to offer a brief summary of our responses and updates:
>
> - Clarification on the novelty and contributions and expanded comparisons with more state-of-the-art methods.
> - Clarification and discussion on the mechanism of the optimization function.
> - discussion on the differences between DISAM and multi-opjective optimization.
>
> **Would you mind checking our responses and confirming if you have any additional questions? We welcome any further comments and discussions!**
>
> Best Regards,
>
> The authors of Submission 1547

---

> > ### Author Response · Authors · 2023-11-23
> > **We anticipate your feedback!**
> >
> > Dear Reviewer sVwJ,
> >
> > The authors greatly appreciate your time and effort in reviewing this submission, and eagerly await your response. We understand you might be quite busy. However, the discussion deadline is approaching, and we have only a few hours left.
> >
> > We have provided detailed responses to every one of your concerns/questions. Please help us to review our responses once again and kindly let us know whether they fully or partially address your concerns and if our explanations are in the right direction.
> >
> > Best Regards,
> >
> > The authors of Submission 1547

---

> ### Comment · Reviewer_d1EH · 2023-11-22
> **Response to the author's rebuttal**
>
> Most concerns were resolved by the author's rebuttal. I really appreciate the author's efforts. However, i have further questions about Q3. To validate the efficacy of DISAM over SAM on incremental application of existing domain generalization methods, the author should also conduct the experiments on
>
> * V-REX+SAM vs V-REX+DISAM
> * V-REX+SAGM vs V-REX+DISAM
>
> and
>
> * Fishr+SAM vs Fishr+DISAM
> * Fishr+SAGM vs Fishr+DISAM
>
> If these experiments (It is okay with simpler evaluation) also shows statistically significant results, i am willing to increase the score from 5 to 6.

---

> > ### Author Response · Authors · 2023-11-22
> > **Further Responses to Reviewer d1EH**
> >
> > **Thanks for your valuable question!** Considering the time-consuming nature of experiments on DomainNet, we have conducted additional experiments on the remaining four datasets in DomainBed (PACS, VLCS, OfficeHome and TerraInc).
> >
> > | **Method**           | **PACS** | **VLCS** | **OfficeHome** | **TerraInc** | **Avg.** |
> > | -------------------- |:--------:|:--------:|:--------------:|:------------:|:--------:|
> > | ERM                  |   85.5   |   77.3   |      66.5      |     46.1     |   68.9   |
> > | SAM                  |   85.8   |   79.4   |      69.6      |     43.3     |   69.6   |
> > | DISAM                |   87.3   |   80.1   |      70.7      |     47.9     |   71.5   |
> > | SAGM                 |   86.6   |   80.0   |      70.1      |     48.8     |   71.4   |
> > | SAGM+DISAM           |   87.5   |   80.7   |      71.0      |     50.0     |   72.3   |
> > | V-REx[1]             |   84.9   |   78.3   |      66.4      |     46.4     |   69.0   |
> > | *V-REx+SAM*          |   86.0   |   77.9   |      68.0      |     45.1     |   69.3   |
> > | **V-REx+DISAM**      |   85.8   |   78.4   |      70.5      |     45.9     |   70.2   |
> > | *V-REx+SAGM*         |   86.1   |   78.4   |      69.6      |     45.4     |   69.9   |
> > | Fishr[2]             |   86.9   |   78.2   |      68.2      |     53.6     |   71.7   |
> > | *Fishr+SAM*          |   87.0   |   78.7   |      69.0      |     47.1     |   70.5   |
> > | **Fishr+DISAM**      |   87.5   |   79.2   |      70.7      |     54.8     |   73.1   |
> > | *Fishr+SAGM*         |   87.0   |   79.3   |      70.6      |     48.5     |   71.4   |
> >
> > As can be seen from the table above:
> >
> > - DISAM **achieves consistently performance improvements** on top of V-REx/Fishr+SAM and V-REx/Fishr+SAGM.
> > - SAM performs poorly on the TerraInc dataset, leading to a decrease in generalization on top of V-REx and Fishr. **SAGM offers a slight improvement, but its enhancement is not as significant as that achieved by DISAM**.
> >
> > We speculate that:
> >
> > - SAM exhibits poor performance on the TerraInc dataset due to **significant domain shift and convergence inconsistency at the domain level**. SAGM partially mitigates this inconsistency issue by constraining gradient directions. However, DISAM directly addresses domain-level convergence inconsistency, leading to a more substantial performance boost.
> > - It is important to note that **DISAM and SAGM improve the first and second steps of SAM separately, allowing for their combination**. In Table 1 of the main text (page 7), we present the notable gains in generalization achieved by combining DISAM and SAGM. **To better present the performance of DISAM, we are currently conducting experiments with V-REx+SAGM+DISAM and Fishr+SAGM+DISAM. We will update the results as soon as possible.**
> >
> >
> > **References:**
> >
> > [1]. Out-of-distribution generalization via risk extrapolation (rex), ICML2021.
> >
> > [2]. Fishr: Invariant gradient variances for out-of-distribution generalization, ICML2022.

---

> ### Author Response · Authors · 2023-11-23
> **Further Response to Reviewer d1EH with more experiments**
>
> Thank you again for the constructive comments. Now, we **complement more experiments regarding different combinations** to provide a comprehensive comparison. Please refer to the following table for the results. Note that, as DomainNet (600,000 images) are too time consuming for us to finish the training in the remaining time of this phase, we provide the results without DomainNet (we are still running on this dataset and can be provided in the final).
>
> | **Method**           | **PACS** | **VLCS** | **OfficeHome** | **TerraInc** | **Avg.** |
> | -------------------- |:--------:|:--------:|:--------------:|:------------:|:--------:|
> | ERM                  |   85.5   |   77.3   |      66.5      |     46.1     |   68.9   |
> | SAM                  |   85.8   |   79.4   |      69.6      |     43.3     |   69.6   |
> | **DISAM**                |   87.3   |   80.1   |      70.7      |     47.9     |   71.5   |
> | SAGM                 |   86.6   |   80.0   |      70.1      |     48.8     |   71.4   |
> | SAGM+DISAM           |   87.5   |   80.7   |      71.0      |     50.0     |   72.3   |
> | V-REx[1]             |   84.9   |   78.3   |      66.4      |     46.4     |   69.0   |
> | *V-REx+SAM*          |   86.0   |   77.9   |      68.0      |     45.1     |   69.3   |
> | **V-REx+DISAM**      |   85.8   |   78.4   |      70.5      |     45.9     |   70.2   |
> | *V-REx+SAGM*         |   86.1   |   78.4   |      69.6      |     45.4     |   69.9   |
> | **V-REx+SAGM+DISAM** |   86.5   |   79.2   |      71.0      |     46.6     |   70.8   |
> | Fishr[2]             |   86.9   |   78.2   |      68.2      |     53.6     |   71.7   |
> | *Fishr+SAM*          |   87.0   |   78.7   |      69.0      |     47.1     |   70.5   |
> | **Fishr+DISAM**      |   87.5   |   79.2   |      70.7      |     54.8     |   73.1   |
> | *Fishr+SAGM*         |   87.0   |   79.3   |      70.6      |     48.5     |   71.4   |
> | **Fishr+SAGM+DISAM** |   87.8   |   80.1   |      71.2      |     55.3     |   73.6   |
>
> Two conclusions can be made:
>
> - Under the backbone of V-REx, we can find that SAM, DISAM and SAGM improve the performance of the vanilla V-REx by 0.3, 1.2, 0.9 and DISAM performs the best. Besides, **DISAM and SAGM both outperform the vanilla SAM from different designing perspectives in optimization, wherein DISAM's perspective seems to be more effective**. Their combinations (i.e., V-REx+SAGM+DISAM) achieves the best performance compared to the either, which demonstrates their **orthogonality and composability**.
> - Under the backbone of Fishr, we can find that SAM even brings the negative impact on the vanilla Fishr (70.5 v.s. 71.7), and **SAGM cannot remedy the negative impact induced by SAM**, yielding the **overall lower** performance than the vanilla Fishr (71.4 v.s. 71.7). **DISAM (73.1) significantly outperforms SAM and SAGM, and exhibits the similar orthogonality and composability with SAGM**.
>
> Although DISAM shows the consistent superiority over SAM and SAGM under the backbones of V-REx and Fishr, we would like to specially summarize their differences as follows for clarity and for the proper claim.
>
> - The vanilla SAM suffer from impairment under multiple domain discrepancy, while DISAM and SAGM both can alleviate this issue, as to some extent **their improvements in design inherently consider the potential bias** under the two-stage SAM optimization procedure.
> - Differently, SAGM makes the change in the **second stage** of SAM, which considers the direction difference between the perturbed gradient by SAM and the original gradient by SGD during gradient updates, by **narrowing the angle between these two gradient directions**. However, **DISAM focuses on the first stage of SAM, namely, the perturbation direction generation process**, enhancing domain-level convergence consistency to achieve better sharpness estimation and, and consequently, improving generalization. DISAM is more direct to intervene the domain shift problem, while SAGM actually makes the implicit effect.
>
> Overall, we do not intend to critize SAGM but to point out the difference under the scenario of our study. We hope the experiments and analysis can address the reviewer's remaining concerns. Any more comments and advices are welcomed.
>
> Best,
>
> The authors of submission 1547.

---

### Author Response · Authors · 2023-11-17
**General Response by Authors**

## Summary

We thank reviewers for their valuable feedback, and appreciate the great efforts made by all reviewers, ACs, SACs and PCs.
We appreciate that the reviewers have multiple positive impressions of our work, including: (1) focused problem is **a common and significant challenge** (mma9, d1EH); (2) **a novel and reasonable method** (mma9, BeTH, d1EH); (3) **comprehensive experiments** (mma9) with **good results** (BeTH); (4) practical with **little additional computational costs**(d1EH);

We provide a summary of our updates, and for detailed responses, please refer to the feedback of each comment/question point-by-point.

- We meticulously enhance the motivation of our study and included additional comparisons and discussions with recent works in the Introduction (refer to Section 1) and the Related Work (see Appendix A). Furthermore, we have provided an in-depth analysis of Figure 1(b) in Appendix C.5.
- We enrich the analysis and explanations of the DISAM algorithm, providing a detailed explanation of the impact of $\rho$ on both generalization and convergence from both theoretical and experimental perspectives (See Appendix B.3 and C.6). And we improve our notation of equations in the Method (refer to Section 3)
- We conduct extensive experimental evaluations against against V-REx[1] and other new state-of-the-art methods (DAC-SC[2] and Fishr[3]) to compare and integrate them with DISAM. These analyses are detailed in our responses to the reviewers' comments. Moreover, we have expanded the comparison results for convergence speed experiments (see Appendix C.7).

The above updates in the revised draft (including the regular pages and the Appendix) are highlighted in blue color.

We once again express our gratitude to all reviewers for their time and effort devoted to evaluating our work. We eagerly anticipate your further responses and are hopeful for a favorable consideration of our revised manuscript.

**Reference**

[1]. Out-of-distribution generalization via risk extrapolation (rex), ICML2021.

[2]. Decompose, Adjust, Compose: Effective Normalization by Playing with Frequency for Domain Generalization, CVPR2023.

[3]. Fishr: Invariant gradient variances for out-of-distribution generalization, ICML2022.

---

### Author Response · Authors · 2023-11-17
**Anonymous repository about our source code**

Thanks for all reviewer's eforts in reviewing our paper. To avoid concerns about the reproducibility and the detailed setups in our experiments, we open our source code in this anonymous repository: https://anonymous.4open.science/r/DISAM-BF40.

---

### Meta-Review · Area_Chair_GWBK · 2023-12-07

**Metareview:**

This paper proposes an improvement to Sharpness-Aware Minimization (SAM), which is a training method aiming to find flat minima and hence improve domain generalization. It is observed that, in the case of multiple source domains, the convergence of SAM in different domains might not be synchronized, which can bias sharpness estimation toward some domains (i.e., those with high losses).  Named Domain-Inspired Sharpness Aware Minimization (DISAM), the proposed method addresses the issue by imposing a variance minimization constraint on domain losses.  The reviewers generally find the problem interesting, the method novel and reasonable, and the experiments comprehensive and the result good.  There is a slight negative score with low confidence (3) and three positive scores with high confidence (4 or 5).

**Justification For Why Not Higher Score:**

See above

**Justification For Why Not Lower Score:**

See above

---

### Decision · Program_Chairs · 2024-01-16

Accept (poster)